

# High-precision muon decay predictions for ALP searches

Pulak Banerjee[1,2], Antonio Coutinho[3,4], Tim Engel[3,5,6], Andrea Gurgone[7,8],
Adrian Signer[3,5] and Yannick Ulrich[9]

**1** Zhejiang Institute of Modern Physics, Department of Physics,
Zhejiang University, Hangzhou 310027, China
**2** Department of Physics, Indian Institute of Technology Guwahati,
Guwahati-781039, Assam, India
**3** Paul Scherrer Institut, CH-5232 Villigen PSI, Switzerland
**4** IFIC, Universitat de València - CSIC, Parc Científic,
Catedrático José Beltrán, 2, E-46980 Paterna, Spain
**5** Physik-Institut, Universität Zürich, Winterthurerstrasse 190, CH-8057 Zürich, Switzerland
**6** Albert-Ludwigs-Universität Freiburg, Physikalisches Institut,
Hermann-Herder-Straße 3, D-79104 Freiburg, Germany
**7** Dipartimento di Fisica, Università di Pavia, Via Agostino Bassi 6, 27100 Pavia, Italy
**8** INFN, Sezione di Pavia, Via Agostino Bassi 6, 27100 Pavia, Italy
**9** Institute for Particle Physics Phenomenology, Department of Physics,
Durham University, Durham, DH1 3LE, UK

## Abstract

We present an improved theoretical prediction of the positron energy spectrum for the polarised Michel decay $\mu^+ \to e^+ \nu_e \bar{\nu}_\mu$. In addition to the full next-to-next-to-leading order correction of order $\alpha^2$ in the electromagnetic coupling, we include logarithmically enhanced terms at even higher orders. Logarithms due to collinear emission are included at next-to-leading accuracy up to order $\alpha^4$. At the endpoint of the Michel spectrum, soft photon emission results in large logarithms that are resummed up to next-to-next-to-leading logarithmic accuracy. We apply our results in the context of the MEG II and Mu3e experiments to estimate the impact of the theory error on the branching ratio sensitivity for the lepton-flavour-violating decay $\mu^+ \to e^+ X$ of a muon into an axion-like particle $X$.



# 1   Introduction

Muon decays are a sensitive probe to test the Standard Model (SM) and search for new physics. In the SM muons decay exclusively through the charged current interaction mediated by the $W^{\pm}$ boson. At low energies this interaction can be described through an effective theory by a $V-A$ four-fermion contact interaction. The coupling or Wilson coefficient of the corresponding dimension 6 operator, the Fermi constant $G_F$, has been measured with high precision [1] through the Michel decay $\mu^+ \to e^+ \nu_e \bar{\nu}_\mu$. Allowing for heavy particles beyond the Standard Model (BSM), more general contact interactions can be generated. Working with the most general, local, derivative-free and lepton-flavour-conserving four-fermion interaction leads to ten Wilson coefficients [2]. A dedicated experimental effort has been carried out to put limits on the related decay parameters [3, 4].

Unless special care is taken, a generic BSM model does not conserve lepton flavour. This leads to charged lepton-flavour-violating (cLFV) processes. If they are mediated by large-mass BSM particles, these processes can be described by a generalised effective theory, containing cLFV four-fermion operators, as well as cLFV dipole interactions [5–7]. Muon decays play again a dominant role in the search for such effects and the current best limits on $\mu \to e\gamma$ [8], $\mu \to eee$ [9] and $\mu N \to eN$ [10] will be further improved in the coming years [11–14].

In this article we focus on an alternative scenario, whereby cLFV muon decays are triggered by low-mass (pseudo)scalar BSM particles with small couplings to SM particles. This class of bosons is generally referred to as axion-like particles (ALPs) [15, 16]. If such a particle $X$ exists with a mass $m_X$ smaller than the muon mass $M$, a new decay channel $\mu^+ \to e^+ X$ for the muon might exist. The detectable final state depends on the lifetime and the dominant decay mode of $X$ [17, 18]. For the prompt decay $\mu^+ \to e^+ X \to e^+(e^+e^-)$, stringent limits have been obtained by the SINDRUM collaboration [19]. If $m_X < 2m$, where $m$ is the mass of the electron, or if the coupling to electrons is strongly suppressed, $X \to \gamma\gamma$ might be the dominant decay mode. The MEG experiment at the Paul Scherrer Institut (PSI) has recently performed a search for $\mu^+ \to e^+ X \to e^+(\gamma\gamma)$ [20].

But the scenario we are going to investigate is the possibility that $X$ escapes undetected. From an experimental point of view, this corresponds to a two-body decay $\mu^+ \to e^+ + \text{invisible}$. For $m_X$ large enough, a limit on the branching ratio can be obtained by looking for a narrow peak in the positron energy spectrum. TWIST has searched for such a signal [21] and has imposed $\mathcal{O}(10^{-5})$ upper limits on the branching ratio for $13\,\text{MeV} < m_X < 80\,\text{MeV}$. The same investigation has been performed by PIENU [22], with resulting limits at the level of

$10^{-4} - 10^{-5}$ in the mass range $47.8 \, \text{MeV} < m_X < 95.1 \, \text{MeV}$. The forthcoming experiment PIONEER [23,24] is expected to improve such limits by $1 - 2$ orders of magnitude, in a similar mass range as PIENU. Furthermore, the Mu$\chi$e experiment [25] has been proposed to explore the mass range $86 \, \text{MeV} < m_X < 105 \, \text{MeV}$ with a branching ratio sensitivity of $10^{-6} - 10^{-7}$.

From a theoretical point of view, there is a particularly strong motivation to look for nearly massless ALPs as they naturally appear as pseudo-Goldstone bosons. Typical examples are the majoron [26], the familon [27] and the QCD axion [28]. In addition, light ALPs produced in cLFV transitions can be sufficiently stable to serve as viable dark matter candidates [29]. In the context of $\mu^+ \to e^+ X$, such a signal would no longer show up as a bump, but as a deviation in the endpoint of the positron energy spectrum. This entails a much more delicate comparison since both experimental and theoretical uncertainties in this region are much more difficult to control. A possibility to suppress the SM background is to consider highly polarised $\mu^+$, such as those produced at high-intensity surface muon beams at TRIUMF and PSI. Notably, surface muons have a polarisation opposite to the momentum. Thus, when they decay at rest, the rate of Michel positrons is suppressed for $c_\theta \equiv \cos\theta \to 1$, where $\theta$ is the angle of the positron momentum with respect to the beam direction. This has been exploited in [30] to obtain $\mathcal{O}(10^{-6})$ limits on right-handed currents as well as $\mu^+ \to e^+ X$ decays for nearly massless $X$.

This process has received considerable interest from the theory community [31–35]. In [36] an analysis has been presented where available and potentially new $\mu^+ \to e^+ X$ searches are compared to other limits for ALPs coupling to leptons. The potential of MEG II together with a forward detector is also investigated there, as it offers the opportunity to focus the search on right-handed signals in a low-background regime. Studies for the process $\mu^+ \to e^+ X$ have been made for Mu3e [37–39] and COMET [40] as well. Another process worth mentioning, considered long ago by Crystal Box [41] and recently for MEG II [42], is $\mu^+ \to e X \gamma$. In [43–45], on the other hand, one finds a more generic analysis of ALP signatures in cLFV phenomena, where both muonic and tauonic decay modes are entertained. Searches for $\tau \to \ell X$ at Belle II are also expected to play a major part in the ongoing probing of flavour-violating new physics, either via prompt decays of possibly short-lived ALPs or, in the case of long-lived ALPs, via displaced vertices or missing energy [46,47]. These processes are, however, beyond the scope of the present work. We will also not discuss the many constraints stemming from astrophysical or cosmological considerations but refer to [36] for an overview.

The purpose of this article is to revisit the search for ALPs in the $\mu^+ \to e^+ X$ decay. This will be done in the context of the currently available muon beams at PSI and the underlying low-energy theory [48], but also in view of HIMB [49], the future high-intensity muon beamline at PSI. Contrary to previous analyses, we do not restrict ourselves to cases where the impact of the SM background is marginal. Rather, we compute the positron energy spectrum from Michel decay of polarised $\mu^+$ as precisely as possible. To this end, we augment the full next-to-next-to-leading order (NNLO) QED calculation [50,51] with the inclusion of collinear logarithms $L_z \equiv \log(z) \equiv \log(m/M)$ and the resummation of soft logarithms. In particular the latter have a large impact at the endpoint of the spectrum. We extend previous calculations by the resummation of soft lograrithms at the next-to-next-to-leading logarithmic order. This background calculation with its conservatively estimated error is contrasted with the signal. At leading order (LO) the signal is just a delta peak in the energy distribution. We also provide a more realistic next-to-leading order (NLO) calculation of the signal [52] to prepare for a more detailed analysis. These theoretical predictions are combined with a realistic estimate of experimental uncertainties. While this is far from a concrete and detailed experimental analysis to be done in connection with an actual measurement, it will give a good indication of the achievable sensitivity, in particular for small $m_X$. Three scenarios will be considered: in addition to the current MEG II and the Mu3e detector set-up, we also contemplate a hypothetical forward detector. As mentioned above, the latter case is particularly interesting for

right-handed ALPs, because the signal and background positrons tend to be emitted in opposite directions.

In order to achieve our goal, we start in Section 2 presenting our new state-of-the-art computation of the positron energy spectrum for the polarised Michel decay $\mu^+ \to e^+ \nu_e \bar{\nu}_\mu$. Section 3.1 contains the NLO calculation of the signal $\mu^+ \to e^+ X$ for a (pseudo)scalar $X$ with generic couplings to leptons. The reasons why the corresponding calculation of $\mu^+ \to e^+ V$ for a (pseudo)vector $V$ is not performed are discussed in Section 3.2. In Section 4, we estimate the sensitivity on the branching ratio of $\mu^+ \to e^+ X$ for our three experimental scenarios, focusing on the impact of the theory error. Finally, our conclusions are presented in Section 5.

## 2 Standard Model prediction for Michel decay

The Michel decay of the muon is one of the best studied processes in particle physics and has been pivotal in shaping our understanding of perturbative higher-order calculations in quantum field theory. Working in the Fermi theory at leading order in the Fermi coupling $G_F$ and to higher orders in the electromagnetic coupling $\alpha$, the inclusive decay rate has been computed at NLO [53] in the very early days of QED. To match the precision of the experiment, an NNLO computation [54, 55] was required, although the effects of the non-vanishing but small electron mass $m$ were only added ten years later [56]. Very recently, first steps have been taken to even go to the three-loop level [57]. While the matching coefficient $G_F$ does not receive higher-order corrections in pure QED, in the full SM there are electroweak effects which have been computed at the two-loop level [58]. The leading effects beyond the Fermi operator can be found in [59, 60].

While these are impressive calculations, the inclusive decay width is of no use in our case. Instead we need the differential decay rate as a function of the positron energy. Such a calculation is more involved as it goes beyond using the optical theorem and requires the separate evaluation of the virtual and real corrections. There are two classes of processes that need to be considered. First, there is the standard Michel process $\mu^+ \to e^+ \nu_e \bar{\nu}_\mu + \{\gamma\}$ where at $N^n$LO we have to take into account up to $n$ additional photons in the final state. Starting from NNLO we also need to consider the case $\mu^+ \to e^+ \nu_e \bar{\nu}_\mu (e^+ e^-) + \{\gamma\}$. We will call these processes open lepton production. For massless electrons, open lepton production has to be combined with Michel decay in order to obtain a collinear safe quantity, as e.g. done in the computation of the decay width [54, 55] or the muon decay spin asymmetry at NNLO [61]. As we will discuss below, however, we will treat the electrons as massive. Therefore, open lepton production is an independent process that in principle can be separated completely from ordinary Michel muon decay. Nevertheless, depending on the details of the experimental analysis, these contributions might need to be included. If this is the case, the definition of the positron energy spectrum has to specify how such events are taken into account. We will include open lepton production through a fixed-order approach and discuss in Section 2.1 how we treat $\mu^+$ decay events with more than one positron in the final state. For the remaining contributions, it is sufficient to consider the standard Michel decay.

In the following, we describe in detail the contributions we include in the differential Michel decay rate of a polarised $\mu^+$, which can conveniently be written as

$$\frac{1}{\Gamma_0} \frac{d^2\Gamma}{dx\, dc_\theta} = F(x, z) - P\, c_\theta\, G(x, z). \tag{1}$$

For a negatively charged $\mu^-$, the sign of the second term changes. As is customary we normalise the decay rate by the LO width $\Gamma_0 = G_F^2 M^5/(192\,\pi^3)$ and split it into an isotropic and anisotropic part $F$ and $G$, respectively. The dependence of these functions on the positron

energy $E$ is often expressed through the dimensionless variable $x \equiv 2E/M$. For a precise prediction at the energy endpoints $2z \leq x \leq (1 + z^2)$ with $z \equiv m/M$, it is important to keep the electron-mass effects, i.e. $z \neq 0$. The conventions for the polarisation of the muon $P$ are chosen such that for perfectly polarised surface $\mu^+$ entering the target along the $z$-axis, we have $P = -1$. More generally, the polarisation vector for $\mu^+$ polarised along the $z$-axis is $\vec{n} = (0, 0, P)$.

If we are completely inclusive with respect to the emission of additional photons, the functions $F$ and $G$ contain all the required information to characterise the positron dynamics. They can be extracted from the decay rate (1) as

$$F(x,z) = \frac{1}{2} \frac{1}{\Gamma_0} \left( \frac{d\Gamma_+}{dx} + \frac{d\Gamma_-}{dx} \right), \qquad G(x,z) = \frac{-1}{P} \frac{1}{\Gamma_0} \left( \frac{d\Gamma_+}{dx} - \frac{d\Gamma_-}{dx} \right), \qquad (2)$$

with

$$\frac{d\Gamma_+}{dx} = \int_0^1 \frac{d^2\Gamma}{dx\, dc_\theta}\, dc_\theta\,, \qquad \frac{d\Gamma_-}{dx} = \int_{-1}^0 \frac{d^2\Gamma}{dx\, dc_\theta}\, dc_\theta\,. \qquad (3)$$

The perturbative expansions of $F$ and $G$ in $a \equiv (\alpha/\pi)$ read

$$F(x,z) = \sum_{n=0} a^n f_n(x,z), \qquad G(x,z) = \sum_{n=0} a^n g_n(x,z). \qquad (4)$$

Since $z$ is small, it is tempting to try to set $z = 0$. However, starting at order $a$ there are terms involving $L_z$. Hence, it is not possible to naively set $z = 0$. Nevertheless, it is possible to consider so-called massified results, dropping all terms that vanish in the limit $z \to 0$ [62–65]. We will use the notation $\tilde{f}_n$ and $\tilde{g}_n$ for these results. In our conventions we have

$$f_0 = \tilde{f}_0 + \mathcal{O}\left(z^2\right) = x^2(3 - 2x) + \mathcal{O}\left(z^2\right), \qquad (5a)$$

$$g_0 = \tilde{g}_0 + \mathcal{O}\left(z^2\right) = x^2(1 - 2x) + \mathcal{O}\left(z^2\right). \qquad (5b)$$

We use the on-shell scheme for the coupling $\alpha$ as well as the masses $m$ and $M$. Our statements or equations are often equally valid for $F$ and $G$. In this case we use the generic notation $H$ as a placeholder for either $F$ or $G$. Similarly, we use the lowercase letter $h_n$ for either $f_n$ and $g_n$, when referring to the single perturbative terms in (4). We also find it convenient to split $h_n$ into photonic and vacuum polarisation (VP) contribution, as $h_n = h_n^\gamma + h_n^{\mathrm{vp}}$. Since the latter start only at NNLO, we have $h_n = h_n^\gamma$ for $n \leq 1$.

The fixed order NNLO contributions $h_2$ will be discussed in Section 2.1. The following two sections deal with improving the photonic terms beyond NNLO by including additional logarithmically enhanced terms. First, Section 2.2 is dedicated to the collinear logarithms of the positron energy spectrum. For each order in $a$ there can be a single power of $L_z \equiv \log(z)$, i.e. $h_n^\gamma \supset a^n L_z^m$ with $m \leq n$. These logarithms cancel for the inclusive result, but have to be kept under control for the differential decay rate. Second, there are also potentially large logarithms at the endpoint of the positron energy spectrum, induced by soft photon emission. They take the form $L_s \equiv \log(1 + z^2 - x)$ and again, for each order in $a$ there can be a single power of $L_s$, i.e. $h_n^\gamma \supset a^n L_s^m$ with $m \leq n$. Those will be addressed in Section 2.3. In addition to $h_n^\gamma$, the VP contributions have to be considered. Keeping both fermion masses different from zero they contain additional logarithms due to the collinear anomaly [65]. For example, $h_2^{\mathrm{vp}}$ has $L_z^3$ terms. These terms would cancel for an observable that is inclusive with respect to the emission of additional electron-positron pairs, but for the positron energy spectrum they contribute. The VP contributions to the soft logarithms are considered in Section 2.4. Finally, in Section 2.5 we will present our final result of the positron energy spectrum with an error estimate. All results presented here are available at the website [66].

## 2.1 Fixed-order results

The NLO results $h_1$ with full $z$ dependence can be found in [67]. A first NNLO calculation of the energy spectrum has been done with a partly numerical approach [68]. Using the analytic results of the two-loop integrals [69] for the heavy-to-light form factor, the NNLO virtual corrections have been computed in [65] and combined with the real corrections [50] in the fully differential Monte Carlo code McMule [51]. Using this code we can compute the distributions defined in (3) at NNLO and, hence, obtain complete results for $h_2$, including all mass effects.

Starting at $\mathcal{O}(\alpha^2)$, also VP contributions $h_n^{\text{vp}}$ exist. This includes closed electron and muon loops, but also loops with tau leptons and hadronic contributions. The contribution of the latter to $h_2$ have been computed in [70] using a dispersive approach. We follow the approach in [71] and use the hyperspherical integration method [72,73] to compute all VP contributions. For the hadronic part, $h_2^{\text{had}}$, the VP itself is evaluated with alphaQEDc19 [74]. At $\mathcal{O}(\alpha^4)$ and beyond, there are also fermion (and hadron) loop contributions other than VP. However, they play no role in our analysis.

At NNLO and beyond, also open lepton production $\mu^+ \to e^+ \nu_e \bar{\nu}_\mu (e^+ e^-)$ might need to be considered. This leads to events with two positrons (and an electron). With a perfect detector, these events can be identified and discarded from the analysis. However, typically the detectors do not hermetically cover the full solid angle. Hence, to allow for more flexibility in the analysis we also provide the contribution of open lepton production to the positron energy spectrum. For this case a precise prescription of how to treat such events with two positrons in the final state is required. Obviously, the choice is not unique. The important point is that the computation is adapted to the experimental analysis. Throughout this paper, we follow the approach to treat all final-state positrons as independent and include both of them in the decay distribution. Hence, a single such muon decay event can lead to up to two entries in the positron energy spectrum. The process $\mu^+ \to e^+ \nu_e \bar{\nu}_\mu (e^+ e^-)$ is fully known at NLO, including all mass effects [75,76]. Hence, its contribution to $h_n$ is available for $n \leq 3$ and is denoted by $h_n^{ee}$. As we will see though, the numerical impact of $h_n^{ee}$ on the analysis as a whole is rather limited.

For all fixed-order contributions to $F$ and $G$ given in this paper, we use a binning in the positron energy $E$ with 3688 bins in total. The width of the bins is 26 keV for $520 \,\text{keV} \leq E \leq 26 \,\text{MeV}$, 16 keV for the following 1000 bins, i.e. until $E \leq 42 \,\text{MeV}$, 8 keV for the following 1000 bins, i.e. until $E < 50 \,\text{MeV}$, and finally 4 keV for the last bins until $E \leq 52.832 \,\text{MeV}$. The non-uniform binning was chosen to have an accurate and efficient sampling of the endpoint region.

## 2.2 Collinear logarithms

For each order in $\alpha$ there can be a collinear logarithm $L_z = \log(z)$ in the positron energy distribution. These logarithms arise due to final-state near-collinear emission regularised by $m \neq 0$ and they cancel for the inclusive result. In this subsection we focus on the purely photonic part $h_n^\gamma$ and denote its leading (LL) and next-to-leading (NLL) collinear logarithmic contribution by $h_n^{\text{cLL}} \sim a^n L_z^n$ and $h_n^{\text{cNLL}} \sim a^n L_z^{n-1}$ respectively. The formalism to determine LL and NLL terms using the fragmentation function approach has been described in [77,78] and was used to predict $f_2^{\text{cNLL}}$ [78] and $g_2^{\text{cNLL}}$ [79]. In [79] also $f_3^{\text{cLL}}$ and $g_3^{\text{cLL}}$ are given.

Following [77,78] and focusing on the purely photonic corrections, we write the $L_z$ enhanced terms of $h^\gamma$ through a convolution

$$\int_x^1 \frac{dx'}{x'} \hat{h}(x', \mu_f) \mathcal{D}\left(\frac{x}{x'}, \mu_f, m\right) + \mathcal{O}(z) \equiv (\hat{h} \otimes \mathcal{D})(x, \mu_f), \tag{6}$$

where $\mathcal{D}$ is the fragmentation function and $\hat{h}$ is related to the energy distribution of a massless positron. For $h_n^{\text{cLL}}$ ($h_n^{\text{cNLL}}$) we need both functions at LO (NLO).

Starting with the fragmentation function, it is understood that in (6) the factorisation scale is set to $\mu_f = M$. Then all large logarithms $L_z$ are contained in $\mathcal{D}$. They are obtained by starting from the initial condition [78, 80] at $\mu_0 = m$

$$\mathcal{D}(x, \mu_0, m) = \delta(1 - x) + \frac{\bar{\alpha}(\mu_0)}{2\pi} d_1(x, \mu_0, m) + \mathcal{O}(\bar{\alpha}^2),$$

$$d_1(x, \mu_0, m) = \left[ \frac{1 + x^2}{1 - x} \left( \log \frac{\mu_0^2}{m^2} - 2\log(1 - x) - 1 \right) \right]_+, \tag{7}$$

and solving the DGLAP equation

$$\frac{\mathrm{d}\mathcal{D}(x, \mu_f, m)}{\mathrm{d}\log\mu_f^2} = \int_x^1 \frac{\mathrm{d}x'}{x'} P_{ee}(x', \bar{\alpha}(\mu_f)) \mathcal{D}\left( \frac{x}{x'}, \mu_f, m \right). \tag{8}$$

Note that the evolution equations are expressed in terms of the $\overline{\text{MS}}$ coupling $\bar{\alpha}(\mu)$. For cNLL accuracy we need the splitting kernels $P_{ee}(x)$ up to $\bar{\alpha}^2$

$$P_{ee}(x, \bar{\alpha}(\mu_f)) = \frac{\bar{\alpha}(\mu_f)}{2\pi} P_{ee}^{(0)}(x) + \left( \frac{\bar{\alpha}(\mu_f)}{2\pi} \right)^2 P_{ee}^{(1)}(x) + \mathcal{O}(\bar{\alpha}^3). \tag{9}$$

Expressed in terms of harmonic polylogarithms [81, 82] they read

$$P_{ee}^{(0)}(x) = \left[ \frac{1 + x^2}{1 - x} \right]_+, \tag{10}$$

$$P_{ee}^{(1)}(x) = \delta(1 - x)\left( \frac{3}{8} - 3\zeta_2 + 6\zeta_3 \right) - \frac{1 + x^2}{1 - x}\left( 4H_{0,0}(x) + 2H_{0,1}(x) + 4H_{1,0}(x) + 2\zeta_2 \right)$$

$$+ (1 + x)H_{0,0}(x) + 2xH_0(x) - 3x + 2. \tag{11}$$

Ignoring VP contributions, the solution of the DGLAP equation at NLL can be written as

$$\mathcal{D}(x, M, m) = \delta(1 - x) + \frac{a}{2}d_1(x, M, m) + \sum_{n=1}^{\infty} (-aL_z)^n \frac{1}{n!} \left( P_{ee}^{(0)} \right)^{\otimes n}(x)$$

$$+ \sum_{n=1}^{\infty} \frac{a}{2}(-aL_z)^n \left\{ \frac{1}{n!}\left[ d_1 \otimes \left( P_{ee}^{(0)} \right)^{\otimes n} \right](x) + \frac{1}{(n-1)!}\left[ \left( P_{ee}^{(0)} \right)^{\otimes(n-1)} \otimes P_{ee}^{(1)} \right](x) \right\}, \tag{12}$$

where $p^{\otimes n}$ refers to the $n$-fold convolution $p \otimes \cdots \otimes p$.

We now turn to the determination of the input function $\hat{h}$. This is done by requiring that (6) reproduces the correct fixed-order massified result $\tilde{h}$. To get $\hat{h}$ at NLO, we need $\tilde{h}$ at NLO. To obtain the latter we expand the full NLO result $h_1$ from [67] in $z$. Writing the result in terms

of harmonic polylogarithms we find

$$
\begin{aligned}
\tilde{f}_1 = \tilde{f}_0\Big(&-H_{0,1}(x) - 4H_{0,0}(x) - 3H_{1,0}(x) - 2\zeta_2\Big) \\
&- \frac{1}{3}\big(11x - 10x^2 + 5x^3\big) + \Big(\frac{5}{6} + 2x - \frac{5}{2}x^2 + \frac{2}{3}x^3\Big)H_0(x) \\
&+ \big(3x + x^2 - 2x^3\big)H_1(x) \\
&+ \Big[\Big(-\frac{5}{6} - 2x + 4x^2 - \frac{8}{3}x^3\Big) + 2\tilde{f}_0\big(H_0(x) + H_1(x)\big)\Big]L_z,
\end{aligned}
\tag{13}
$$

$$
\begin{aligned}
\tilde{g}_1 = \tilde{g}_0\Big(&-H_{0,1}(x) - 4H_{0,0}(x) - 3H_{1,0}(x) - 2\zeta_2\Big) \\
&- \frac{1}{6}\big(3 - 10x - 13x^2 + 8x^3\big) - \Big(\frac{1}{6} + \frac{7}{2}x^2 - \frac{2}{3}x^3\Big)H_0(x) \\
&+ \Big(\frac{2}{3x} - 2 + 3x - \frac{5}{3}x^2 - 2x^3\Big)H_1(x) \\
&+ \Big[\Big(\frac{1}{6} + 4x^2 - \frac{8}{3}x^3\Big) + 2\tilde{g}_0\big(H_0(x) + H_1(x)\big)\Big]L_z,
\end{aligned}
\tag{14}
$$

with $\tilde{h}_0$ given in (5). With this input we can deduce from (6) the functions $\hat{h}$ as

$$
\hat{h}_0(x) = \tilde{h}_0,
\tag{15}
$$

$$
\hat{h}_1(x, \mu_f) = \tilde{h}_1 + \frac{1}{2}\left(\log\frac{\mu_0^2}{\mu_f^2}P_{ee}^{(0)}(x) - d_1(x, \mu_0, m)\right) \otimes \tilde{h}_0,
\tag{16}
$$

where the explicit factor $1/2$ in (16) appears since $h_n$ is expanded in $a = \alpha/\pi$ and not $\alpha/(2\pi)$. The dependence on the scale $\mu_0$ cancels between the two terms in parenthesis in (16), as expected. The resulting $\log(m^2/\mu_f^2)$ combines with the $L_z$ terms of $\tilde{h}_1$ to logarithms of the form $\log(M^2/\mu_f^2)$. Hence, $\hat{h}$ does not contain large logarithms. The results obtained by (16) agree with those given in [78] for $\hat{f}_1$ and in [79] for $\hat{g}_1$.

With these results and using the Mathematica package MT [83], we can calculate the purely photonic cLL and cNLL contributions, in principle to any order in $\alpha$. In practice, we have stopped at $n = 4$. The results for $h_n^{\text{cLL}}$ and $h_n^{\text{cNLL}}$ for $n \leq 4$ are attached in a ancillary file.

## 2.3 Soft logarithms

In addition to the collinear logarithms mentioned above, at the endpoint $x \to (1+z^2)$, there are also soft logarithms $L_s = \ln(1 + z^2 - x)$. Again, for each order of $\alpha$ there can be a single power of $L_s$. The leading logarithms $(a L_s)^n$ have been considered before [79]. For our purpose, this is not sufficient. We extend these results by including next-to-leading $a(a L_s)^n$ and next-to-next-to-leading $a^2(a L_s)^n$ soft logarithms.

In QED, multiple soft emission follows the Yennie-Frautschi-Suura (YFS) exponentiation [84] which allows for the resummation of leading terms. The corresponding result can be written as

$$
h_{[0^+]}^{\gamma,s} = h_0 \exp\big(a c_s L_s\big) = h_0\big(1 + z^2 - x\big)^{a c_s}.
\tag{17}
$$

We generally use the subscript $n^+$ to denote a contribution that contains terms of order $n$ and higher. Since these contributions are not proportional to a fixed power of $a$ we include the coupling in $h_{[n^+]}^{\gamma,s}$. The meaning of the square brackets in the subscript will be explained after (18). In (17) the coefficient $c_s$ can be obtained as the coefficient of the $L_s$ term of $f_1$

or $g_1$ after taking the limit $x \to (1 + z^2)$. Alternatively, the fragmentation function approach can be considered in the soft limit. In this limit it is possible to solve the evolution equations analytically [85–87]. For the leading soft logarithms, i.e. for $h^{\gamma,s}_{[0^+]}$ the two approaches yield the same results, as required, namely

$$c_s = 2 \frac{1 - z^2 + (1 + z^2)L_z}{(z^2 - 1)} \simeq -2(1 + L_z). \tag{18}$$

The nature of the soft logarithms is different from the collinear logarithms in that they are not simply large but actually divergent. Indeed, each perturbative power in (17) is ill defined at the endpoint $x = (1 + z^2)$ and only after integrating over (arbitrarily small) bin sizes in the energy (or in $x$) a finite result is obtained. On the other hand, after resummation the expression is mathematically well defined even at the endpoint, since $c_s \geq 0$. To indicate that $h^s_{[0^+]}$ is pointwise finite we use brackets in the subscript. In fact, the functions $h_{[n^+]}$ are not only pointwise finite but also tend to 0 at the endpoint, albeit often very sharply.

Beyond NLO there are also further suppressed soft logarithms $a^n h_n \supset a^n L^j_s$ with $j < n$. To obtain them, we use

$$h^{\gamma,s}_{[1^+]} = h_0 \, a \, k^\gamma_1 \left(1 + z^2 - x\right)^{a \, c_s}, \tag{19a}$$

$$h^{\gamma,s}_{[2^+]} = h_0 \, a^2 \, k^\gamma_2 \left(1 + z^2 - x\right)^{a \, c_s}, \tag{19b}$$

$$h^{\gamma,s}_{[3^+]} = h_0 \, a^3 \, k^\gamma_3 \left(1 + z^2 - x\right)^{a \, c_s}, \tag{19c}$$

where again we consider only purely photonic corrections. As before, $h^{\gamma,s}_{[n^+]}$ contains not only terms $a^n$ but also higher powers of $a$. To determine the coefficients $k^\gamma_i$ we take the limit $x \to 1 + z^2$ of the analytic result for $f^\gamma_i / f_0$ or $g^\gamma_i / g_0$. Neglecting terms suppressed by $z$ for $k^\gamma_1$ this yields

$$\frac{h^\gamma_1}{h_0} \to -2 - \frac{3}{2}L_z - 2(1 + L_z)L_s \qquad \Longrightarrow \qquad k^\gamma_1 = -2 - \frac{3}{2}L_z. \tag{20}$$

The $L_s$ term in (20) is reproduced through (17) and (18), while the terms involving $k^\gamma_1$ of (19a) are of the form $a^n L^{n-1}_s$ and correspond to the next-to-leading soft logarithms. Proceeding along the same way for $h^\gamma_2$ we can determine $k^\gamma_2$ that is required for the next-to-next-to-leading soft logarithms of the form $a^n L^{n-2}_s$. However, since the analytic result for $h^\gamma_2$ is not known, we use the cNLL approximation to $h^\gamma_2$ to obtain the analytic form of the $L_z$ terms and use a fit to the numerical result $h^\gamma_2$ for the constant term. Thus, we find

$$k^\gamma_2 = L^2_z \left(\frac{9}{8} - 2\zeta_2\right) + L_z \left(\frac{45}{16} - \frac{5}{2}\zeta_2 - 3\zeta_3\right) + k^\gamma_{2,0}, \tag{21}$$

with $k^\gamma_{2,0} = -6 \pm 1$. Since the impact of the rather large error in the extraction of $k^\gamma_{2,0}$ has no adverse effect on the analysis as a whole, no particular effort has been made to improve the precision. Again, the coefficients in (21) also contain terms suppressed by $z$ but their numerical impact is tiny. Combining (20) and (21) with (19a) and (19b), respectively, reproduces all next-to-leading $a^n L^{n-1}_s$ and next-to-next-to-leading $a^n L^{n-2}_s$ soft logarithms in $h^\gamma_n$. Similarly we can use $h^{cLL}_3$ and $h^{cNLL}_3$ to obtain

$$k^\gamma_3 = L^3_z \left(-\frac{9}{16} + 3\zeta_2 - \frac{8}{3}\zeta_3\right) + L^2_z \left(-\frac{63}{32} + \frac{31}{4}\zeta_2 - \frac{7}{2}\zeta_3\right) + \dots, \tag{22}$$

where terms $\sim L_z$ and terms without $L_z$ are not known. Using (22) in (19c) reproduces all terms $a^n L^{n-3}_s L^n_z$ and $a^n L^{n-3}_s L^{n-1}_z$ in $a^n h^\gamma_n$ but misses terms $a^n L^{n-3}_s L^m_z$ with $m \leq n-2$. We have

verified that the soft logarithms $L_s$ produced through (19) are consistent with the $\log(1-x)$ terms of $h_4^{\text{cLL}}$ and $h_4^{\text{cNLL}}$.

Since the soft logarithms $L_s$ are included in (17) and (19), we can subtract them from the fixed-order results $h_n^\gamma$ and their collinear approximations $h_n^{\text{cLL}}$ and $h_n^{\text{cNLL}}$. We will denote these subtracted results by additional brackets in the subscript to indicate they are pointwise finite. Formally, we write

$$h_{[n]}^\gamma \equiv h_n^\gamma - \left( \sum_{j \leq n} h_{[j^+]}^{\gamma,\text{s}} \right)\Bigg|_{a^n \text{ coeff.}} = h_n^\gamma - h_0 \sum_{i=0}^{n} k_{n-i}^\gamma \frac{c_s^i L_s^i}{i!} \,. \tag{23}$$

Note $h_{[0]}^\gamma = 0$, since the full tree-level result is contained in $h_{[0^+]}^{\gamma,\text{s}}$. Since we also include the term $j = n$ or $i = 0$ on the r.h.s. of (23), the functions $h_{[n]}^\gamma$ are equal to zero at the endpoint.

Similarly, we define $h_{[n]}^{\text{cLL}}$ and $h_{[n]}^{\text{cNLL}}$. In this case only the leading or next-to-leading $L_z$ terms are subtracted. Also, we adapt the subtraction terms by setting $z = 0$ in the coefficients and in $L_s$.

## 2.4 Vacuum polarisation contributions

As mentioned in Section 2.1 at NNLO also VP terms $h_2^{\text{vp}}$ are included in $h_2$. Electron loops are by far the dominant VP contributions and they also cause the collinear anomaly that is an additional source of logarithms of the form $\ln\left(mM/(2ME)\right) = \ln(z/x)$ [65]. For large $x$ they have the same form as collinear logarithms. Thus, in order to take into account the dominant effect of electron loops beyond NNLO, we consider their contribution analogous to (19) and define

$$h_{[2^+]}^{\text{vp,s}} = h_0 \, a^2 k_2^{\text{vp}}\left(1 + z^2 - x\right)^{a \, c_s} \,. \tag{24}$$

The coefficient $k_2^{\text{vp}}$ can be computed analytically by considering the limit $x \to 1 + z^2$ of the two-loop contribution with an electron (and muon) loop. We find

$$\begin{aligned}
k_2^{\text{vp}} = &\, n_e \left( \frac{535}{108} + \frac{11}{18}\zeta_2 + \frac{1}{3}\zeta_3 + \left[ \frac{397}{108} + \frac{2}{3}\zeta_2 \right]L_z + \frac{25}{18}L_z^2 + \frac{2}{9}L_z^3 \right) \\
&+ n_\mu \left( \frac{12991}{1296} - \frac{53}{9}\zeta_2 - \frac{1}{3}\zeta_3 \right),
\end{aligned} \tag{25}$$

where we have labelled the contribution of the electron and muon through the bookkeeping parameters $n_e = 1$ and $n_\mu = 1$, respectively. The hadronic and tau-loop contributions are neglected in $k_2^{\text{vp}}$. In contrast to the collinear logarithms discussed in Section 2.2, here we get up to three powers of $L_z$ at order $\alpha^2$. This triple logarithm is cancelled by the open lepton production in inclusive quantities. The appearance of a triple logarithm in the real NNLO contribution can be understood by noting that the mass of the electron does not only regularise the collinear singularities, but also prevents the intermediate photon to become soft. This is consistent with the observation that integration over the phase space of $\gamma^* \to q\bar{q}q'\bar{q}'$ yields poles $1/\epsilon^3$ in the double-real part for two-jet production [88,89].

In analogy to (23) we define

$$h_{[2]}^{\text{vp}} \equiv h_2^{\text{vp}} - h_{[2^+]}^{\text{vp,s}}\Big|_{a^2 \text{ coeff.}} = h_2^{\text{vp}} - a^2 h_0 \, k_2^{\text{vp}} \,, \tag{26}$$

as all VP terms of order $\alpha^2$ that are not included in (24). This is a slight abuse of our notation, as $h_2^{\text{vp}}$ is already pointwise finite. On the other hand, $h_{[2]}^{\text{vp}}$ is not only finite but actually vanishes at the end point $x \to 1 + z^2$. We stress that in $h_{[2]}^{\text{vp}}$ hadronic and tau loops are included even though their numerical impact is limited.

## 2.5 Final result and theory error

With all these results at hand, we define our best prediction for the positron energy spectrum as

$$
\begin{aligned}
H = h^{\mathrm{s}}_{[0^+]} + \left( h^{\gamma,\mathrm{s}}_{[1^+]} + a\, h^{\gamma}_{[1]} \right) + \left( h^{\gamma,\mathrm{s}}_{[2^+]} + a^2\, h^{\gamma}_{[2]} \right) + \left( h^{\mathrm{vp,s}}_{[2^+]} + a^2\, h^{\mathrm{vp}}_{[2]} \right) \\
+ a^3 \left( h^{\mathrm{cLL}}_{[3]} + h^{\mathrm{cNLL}}_{[3]} \right) + a^2\, h^{ee}_2 \, .
\end{aligned}
\tag{27}
$$

The first term, $h^{\mathrm{s}}_{[0^+]}$, contains all tree-level terms as well as the leading soft logarithms at all orders in $a$. The second term, $h^{\gamma,\mathrm{s}}_{[1^+]} + a\, h^{\gamma}_{[1]}$, contains all terms at order $a$, except the leading soft logarithm to avoid double counting. In addition, it includes the next-to-leading soft logarithms at all orders in $a$. Similarly, $h^{\gamma,\mathrm{s}}_{[2^+]} + a^2\, h^{\gamma}_{[2]}$ has all terms at order $a^2$ except those already included in $h^{\gamma,\mathrm{s}}_{[n^+]}$ with $n \leq 1$, but augmented by the next-to-next-to-leading soft logarithms at all orders in $a$. The leading and next-to-leading collinear logarithms at order $a^3$ that are not part of $h^{\gamma,\mathrm{s}}_{[n^+]}$ with $n \leq 2$ are included in $h^{\mathrm{cLL}}_{[3]}$ and $h^{\mathrm{cNLL}}_{[3]}$, respectively. We recall that neither $h^{\mathrm{cLL}}_{[3]}$, $h^{\mathrm{cNLL}}_{[3]}$ nor $h^{\gamma,\mathrm{s}}_{[n^+]}$ and $h^{\gamma}_{[n]}$ contain VP contributions. The latter are taken into account separately in $h^{\mathrm{vp,s}}_{[2^+]}$ and $h^{\mathrm{vp}}_{[2]}$. In (27) the contribution from open lepton production, $h^{ee}_2$, is listed separately, as it might or might not be included.

Our final result for $F$ according to (27), omitting $f^{ee}_2$, is shown in the top panel of Figure 1 (orange) compared to the tree-level result $f_0$ (green). The individual contributions of (27) are depicted in the various sub-panels. We note a nice convergence in that the successive terms $f^x_{[n]}$ are suppressed by $10^{-2n}$ with respect to $f_0$. As indicated in the lowest two panels, this also holds for $f^{\mathrm{cLL}}_{[4]}$ and $f^{\mathrm{cNLL}}_{[4]}$ as well as the approximate $f^{\gamma,\mathrm{s}}_{[3^+]}$ and $f^{\gamma}_{[3]}$ which are not included in (27). The open lepton contribution $f^{ee}_2$ shown in the middle panel has a relative effect smaller than $10^{-5}$ on the distribution for energies larger than $\sim 40\,\mathrm{MeV}$.

The picture is similar for $G$, depicted in Figure 2. One notable difference is that the collinear logarithms are numerically more important near the endpoint of the spectrum. This will affect the theoretical error to which we turn now.

The error is evaluated as

$$
\delta H = \delta h^{\mathrm{MC}} \oplus \delta h^{ee} \oplus \sqrt{2} \max \left\{ \delta h^{\mathrm{c}}, \delta h^{\mathrm{s}}, \delta h^{\mathrm{had}}_2, \delta h^{\mathrm{vp}}, \delta h^{\mathrm{EW}} \right\},
\tag{28}
$$

where $\oplus$ indicates we add the errors in quadrature. The first term, $\delta h^{\mathrm{MC}}$ is the numerical error of the Monte Carlo. The second term, $\delta h^{ee}$, is the error induced through open lepton production, assuming these events are included according to the prescription described in Section 2.1. Finally, the various terms in the curly brackets correspond to theoretical errors due to imperfect calculations of the Michel decay. As we will see, there is typically a single dominant term. Hence, there is little difference in whether we take the maximum, or add these errors linearly or in quadrature. We have decided to take the maximum, but multiplied by a factor $\sqrt{2}$ to have a conservative estimate also in the case when there are two error contributions of similar size. The individual terms of (28) are depicted in Figure 3 for $F$ and $G$, respectively and will be discussed in what follows.

Starting with $\delta h^{\mathrm{MC}}$, the numerical error of the Monte Carlo, we note that it is completely dominated by the numerical error of the NNLO corrections. Hence, we set $\delta h^{\mathrm{MC}} = \delta h^{\mathrm{MC}}_2$. In principle this error can be reduced by increasing the statistics. However, for the binning we have chosen in practice it is difficult (and not necessary) to obtain an error much smaller than $\delta h^{\mathrm{MC}}_2 = 5 \times 10^{-7}\, h_0$ for $E < 49\,\mathrm{MeV}$ and $\delta h^{\mathrm{MC}}_2 = 3 \times 10^{-6}\, h_0$ for $E \geq 49\,\mathrm{MeV}$.

Turning to $\delta h^{ee}$, we are in the comfortable position that open lepton production is a rather small effect anyway and the NLO corrections which contribute at $\alpha^3$ to $h^{ee}$ are known [75]. The additional suppression by a factor of $a$ renders $h^{ee}_3$ very small and we can afford to take

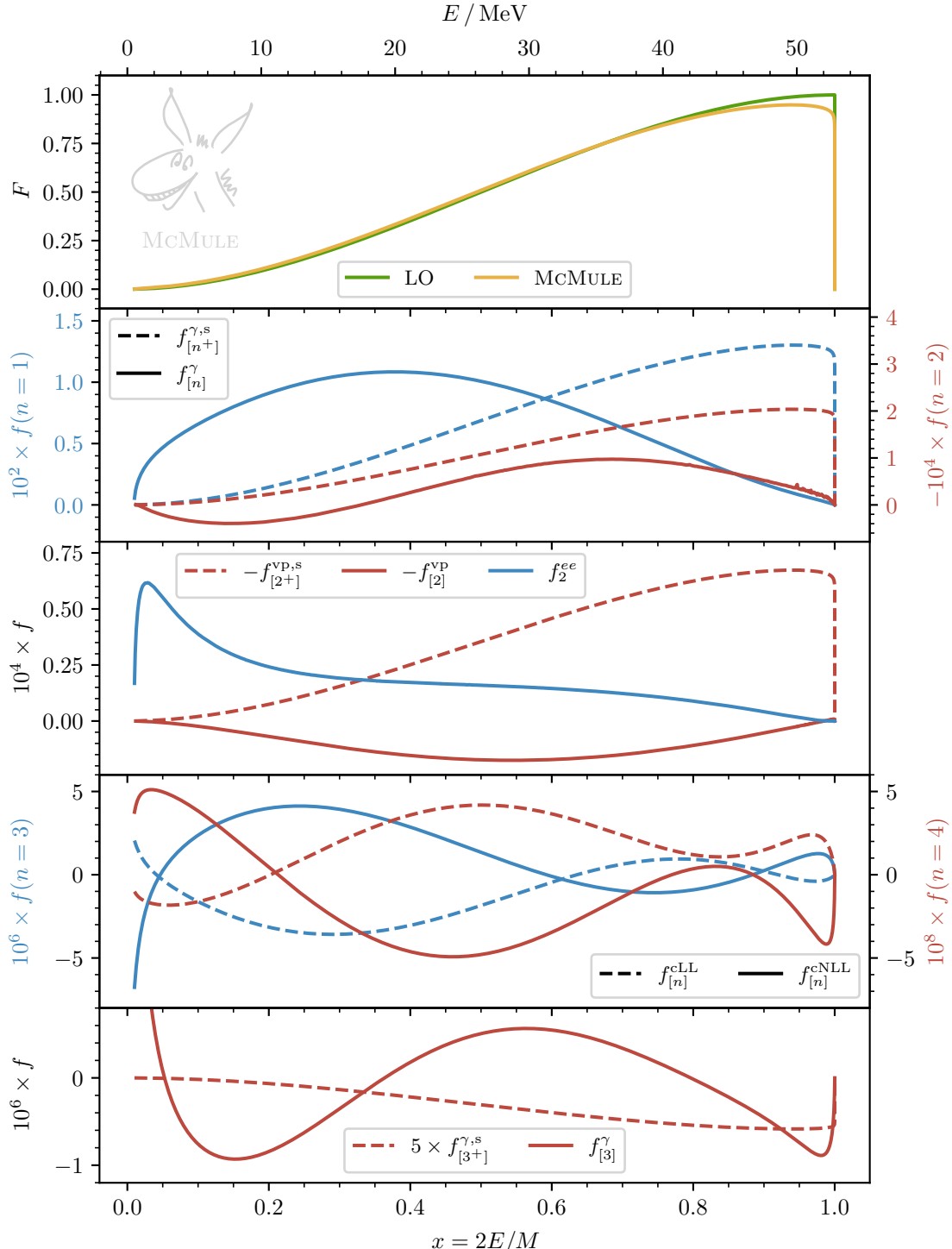

Figure 1: The best theory prediction for $F$ according to (27). The top most panel contains $f_0$ (green) and $F$ (orange). In the next panels we show the most important individual contributions listed in (27), as well as $f_{[4]}^{\text{cLL}}$, $f_{[4]}^{\text{cNLL}}$, $f_{[3^+]}^{\gamma,\text{s}}$, and $f_{[3]}^{\gamma}$ which are used for the error estimate.

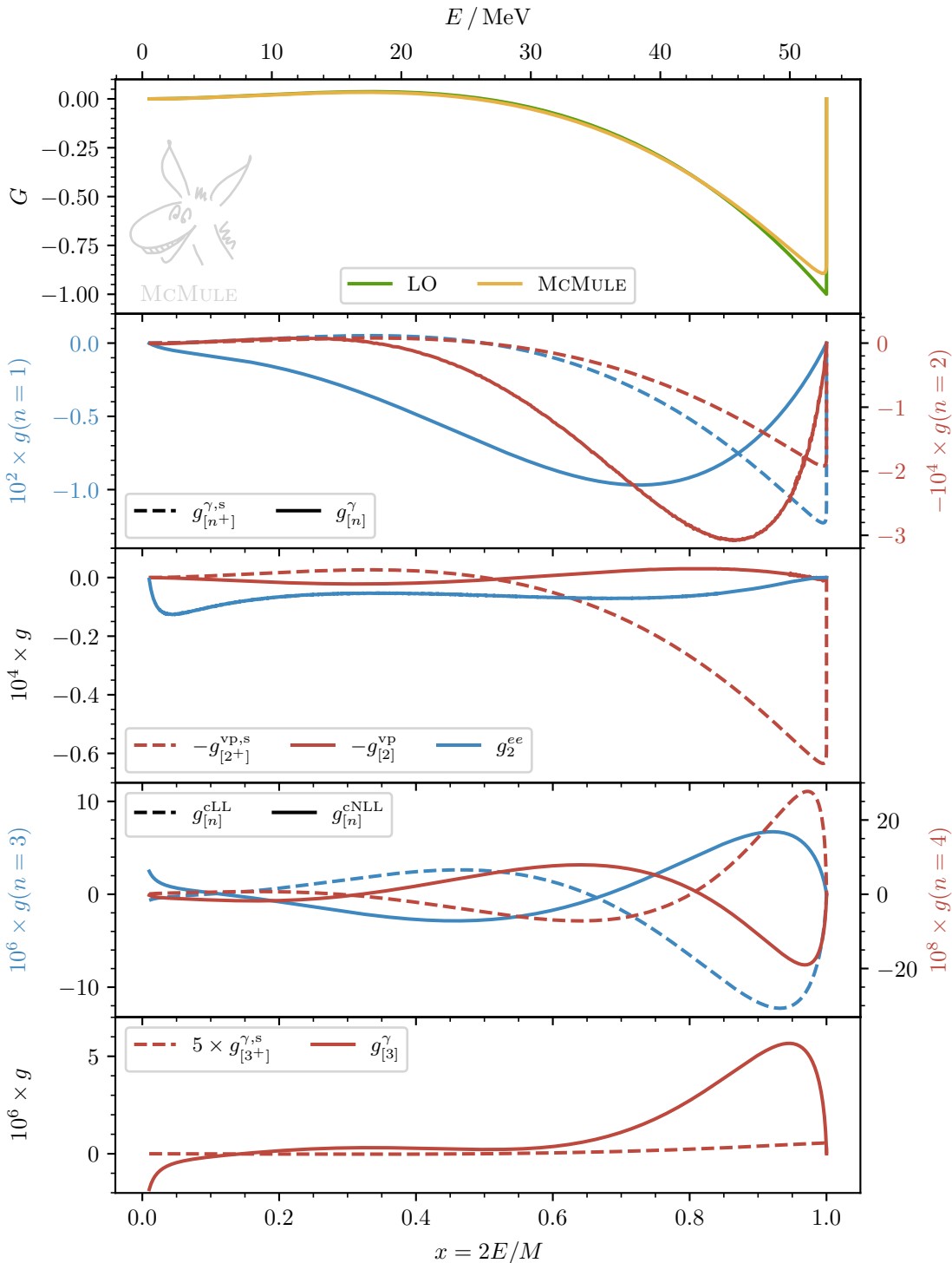

Figure 2: The best theory prediction for $G$ according to (27). The top most panel contains $g_0$ (green) and $G$ (orange). In the next panels, we show the most important individual contributions listed in (27), as well as $g_{[4]}^{\text{cLL}}$, $g_{[4]}^{\text{cNLL}}$, $g_{[3+]}^{\gamma,\text{s}}$, and $g_{[3]}^{\gamma}$ which are used for the error estimate.

this last known correction $\delta h^{ee} = h_3^{ee}$ as a conservative error estimate. The smallness of $\delta h^{ee}$ also implies that the details of how to treat decays with more than one positron in the final state do not affect the main conclusions.

This leaves us with the most delicate case, a reliable error estimate due to missing corrections in the Michel decay. As discussed in the previous subsections, this includes an error $\delta h^c$ due missing collinear logarithms, an error $\delta h^s$ due to missing terms in the soft logarithms, and errors $\delta h_2^{had}$ and $\delta h^{vp}$ due to imperfect knowledge of vacuum polarisation contributions. For completeness, we also include an error $\delta h^{EW}$ due to our neglect of electroweak terms beyond the Fermi theory. Our general strategy to estimate the error due to missing higher-order terms is to take the last term in the perturbative expansion that can be reliably computed.

As mentioned above, all terms up to $\mathcal{O}(\alpha^2)$ are taken into account in (27). Considering the collinear logarithms beyond this order, their contributions to $G$ are not converging quite as well as the other higher-order in $\alpha$ terms. Hence, we take the very conservative approach to assign an error corresponding to the last term that is included in the final result. Concretely, we associate an error that is equal to the collinear logarithms of order $\alpha^3$, i.e we set $\delta h^c = |h_{[3]}^{cLL} + h_{[3]}^{cNLL}|$. From the fourth panel of Figure 1 and Figure 2 we see that even higher order collinear logarithms $h_{[4]}^{cLL}$ and $h_{[4]}^{cNLL}$ are considerably smaller.

Moving to the soft logarithms, two components contribute to the error $\delta h^s = |h_{[3^+]}^{\gamma,s}| + |\delta h_{[2^+]}^{\gamma,s}|$. The first term $h_{[3^+]}^{\gamma,s}$ is evaluated using (19c) and (22). It corresponds to those soft logarithms beyond next-to-next-to-leading logarithmic accuracy that are enhanced by the maximal and next- to-maximal power of $L_z$. Hence, this is a reliable estimate for the neglect of terms beyond $\delta h_{[2^+]}^{\gamma,s}$ in (27). The second term $\delta h_{[2^+]}^{\gamma,s}$ has a numerical origin and is induced by the error of the fitted coefficient $k_{2,0}^{\gamma} = -6 \pm 1$.

The error $\delta h_2^{had}$ is induced by imperfect knowledge of the hadronic VP. Taking a very conservative approach and assuming a flat (independent of the kinematics) 5% uncertainty, we assign an error $\delta h_2^{had} = 0.05 \cdot |h_2^{had}|$.

The error $\delta h^{vp}$ associated with the VP contribution also consists of two parts. First, we estimate the effect of missing multiple insertions of electron loop effects by the difference from using the on-shell coupling and the $\overline{\text{MS}}$ coupling. This yields $\delta h_{[2^+]}^{vp} = 2/3 \, a L_z \, h_{[2^+]}^{vp} \simeq a L_z \, h_{[2^+]}^{vp}$. Second, we estimate the effect of missing higher-order corrections in the VP itself. To this end, we insert the two-loop result of the electron VP, which can be extracted from [90], in the computation of the positron energy spectrum. Denoting this result by $\delta h_3^{vp}$, the total VP error is then taken to be $\delta h^{vp} = |\delta h_{[2^+]}^{vp}| + |\delta h_3^{vp}|$.

Turning to $\delta h^{EW}$, the leading corrections beyond the Fermi theory are $h^{EW} = h_0 \, 3M^2/(5M_W^2)$. Their relative numerical impact is of the order of $10^{-6}$ and sufficiently small to allow us to take $\delta h^{EW} = h^{EW}$.

There are further tiny contributions that have not been considered in (27) and (28), such as $z$ suppressed effects beyond NNLO. However, they can be safely ignored.

To summarise, for the individual contributions in (28) we choose

$$
\begin{aligned}
\delta h^{ee} &= h_3^{ee}\,, \\
\delta h^c &= |h_{[3]}^{cLL} + h_{[3]}^{cNLL}|\,, \\
\delta h^s &= |h_{[3^+]}^{\gamma,s}| + |\delta h_{[2^+]}^{\gamma,s}|\,, \\
\delta h^{had} &= 5\% \times h_2^{had}\,, \\
\delta h^{vp} &= a|L_z \, h_{[2^+]}^{vp}| + |\delta h_3^{vp}|\,, \\
\delta h^{EW} &= h^{EW}\,.
\end{aligned}
\tag{29}
$$

The total relative error $\delta H/H$ and its individual contributions are depicted in Figure 3. In the region we are interested in, the dominant errors are from the missing soft logarithms

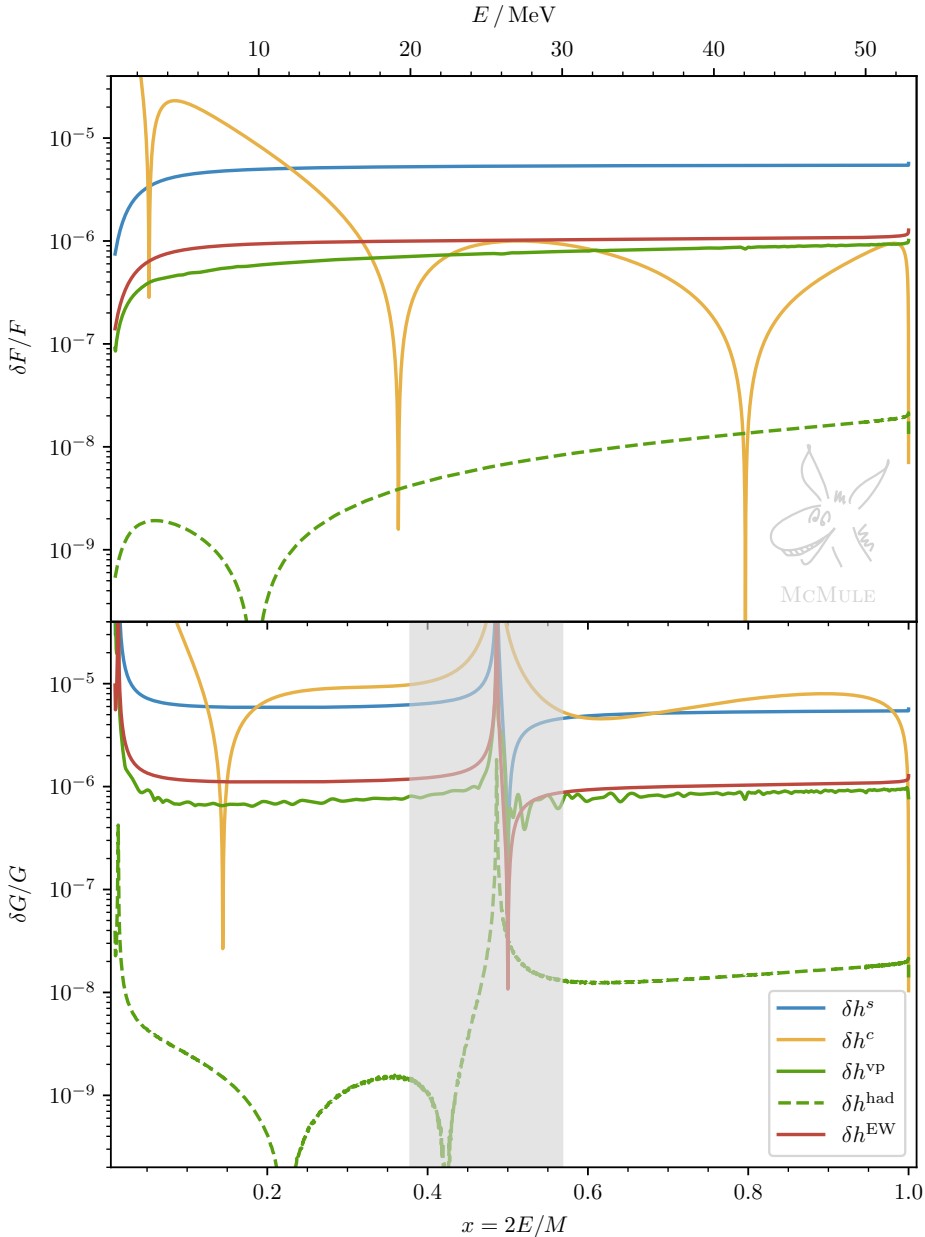

Figure 3: The errors $\delta F/F$ (top panel) and $\delta G/G$ (lower panel) as defined in (28). The downward spikes in $\delta h^{\mathrm{c}}$ (orange curves) appear due to zeros in the numerator. For $x \simeq 0.5$ the denominator $G$ is zero, resulting in an artificial enhancement. This region is indicated by a grey band. Not shown is the Monte Carlo error $\delta h^{\mathrm{MC}}/H$ which increases to $3 \times 10^{-6}$ for large $x$.

and in the case of $G$ also from the missing collinear logarithms. The downward peaks of some contributions are due to zero crossings. Since $\delta G/G$ formally diverges at the zero crossing of $G$, the energy range $20\,\text{MeV} < E < 30\,\text{MeV}$ is shown behind a grey band in Figure 3.

To conclude, we note that near the endpoint of the positron energy spectrum, the relative precision is $\delta F/F \lesssim 5 \times 10^{-6}$ and the dominant error is from the missing terms in the soft resummation. The purely numerical error from the Monte Carlo integration is of a similar size as is the contribution from open lepton production. For $G$ we have $\delta G/G \lesssim 10^{-5}$ and the missing collinear logarithms are roughly as important as the missing soft logarithms.

# 3 LFV muon decay in simplified models

In this section we discuss the calculation of the signal $\mu \to eX$. At tree-level we have two particles with fixed energy in the final state. A more realistic description of the final state is obtained by including QED corrections at NLO, which have the main effect of adding a radiative tail to the peak. This will be done in Section 3.1 for a generic scalar particle $X$. Some remarks regarding a vector particle are made in Section 3.2, where we also argue that a separate analysis is not required.

## 3.1 LFV scalar particles

The coupling of ALPs to leptons is often written as a dimension 5 operator with a derivative coupling, divided by a large scale $\Lambda$. The Lagrangian reads

$$\mathcal{L}_X = \frac{1}{\Lambda}\left(\partial_\mu X\right)\bar{\psi}_j\left(g_V^{jl}\gamma^\mu + g_A^{jl}\gamma^\mu\gamma^5\right)\psi_l\,, \tag{30}$$

where the family indices $1 \le j, l \le 3$. For ALPs that are pseudo-Goldstone bosons, $\Lambda$ corresponds to the scale of spontaneous symmetry breaking and the vector coupling $g_V = 0$. Assuming anomaly-free vector and axial currents, the derivative coupling can be expressed via integration by parts in terms of the Yukawa-like couplings [91]

$$\mathcal{L}_X = -\frac{i}{\Lambda}X\,\bar{\psi}_j\left[g_V^{jl}(m_j - m_l) + g_A^{jl}(m_j + m_l)\gamma^5\right]\psi_l\,. \tag{31}$$

Here, we take a bottom-up approach and simply investigate a scalar particle $X$ of mass $m_X$ that couples to leptons through

$$\mathcal{L}_X = X\,\bar{\psi}_j\left(C_L^{jl}P_L + C_R^{jl}P_R\right)\psi_l\,, \tag{32}$$

without further specifying the nature of $X$. The operators $P_L = (1-\gamma^5)/2$ and $P_R = (1+\gamma^5)/2$ are used to project on the left- and right-handed parts. The coupling matrices satisfy $C_L^\dagger = C_R$. Since we are dealing with the decay $\mu^+ \to e^+X$ we set the family indices to $j = 2$ and $l = 1$ and use a short-hand notation $C_L \equiv C_L^{21}$ and $C_R \equiv C_R^{21}$. These couplings are related to $g_V \equiv g_V^{21}$ and $g_A \equiv g_A^{21}$ of (30) as

$$C_L = g_V\frac{i(m-M)}{\Lambda} + g_A\frac{i(M+m)}{\Lambda}\,, \qquad C_R = g_V\frac{i(m-M)}{\Lambda} - g_A\frac{i(M+m)}{\Lambda}\,. \tag{33}$$

For $g_V = g_A$ we obtain a right-handed $V+A$ coupling, while for $g_V = -g_A$ we get a left-handed $V-A$ coupling. If $g_A = 0$ the ALP is purely scalar ($V$), while it is pseudoscalar ($A$) if $g_V = 0$. In the following we will usually refer to these four typical chiral structures.

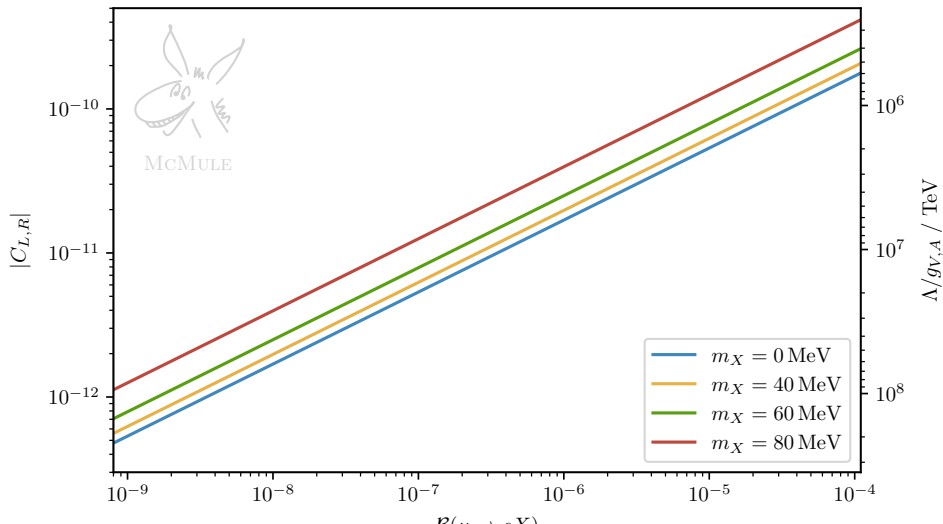

Figure 4: Conversion plot between the LO branching ratio $\Gamma_0^X/\Gamma_0$ and the couplings $C_{L,R}$.

At LO, in the rest frame of the muon, the decay $\mu^+ \to e^+ X$ results in a back-to-back $e^+$-$X$ pair with momentum $|\vec{q}| = \sqrt{\lambda}/(2M)$ expressed through the usual Källén function $\lambda \equiv \lambda(M^2, m^2, m_X^2)$. This results in positrons of energy

$$E_X = \frac{M^2 + m^2 - m_X^2}{2M}, \tag{34}$$

and a partial decay rate [34]

$$\Gamma_0^X \equiv \Gamma_0(\mu \to eX) = \frac{|\vec{q}|}{16\pi}\Big[(1 + z^2 - r^2)\big(|C_L|^2 + |C_R|^2\big) + 4z\,\mathrm{Re}\big(C_R C_L^*\big)\Big], \tag{35}$$

where we have defined $r \equiv m_X/M$. Since MEG II can detect positrons with $E \gtrsim 45$ MeV, the experiment is sensitive to signals with $m_X \lesssim 40$ MeV. Similarly, Mu3e can detect positrons with $E \gtrsim 10$ MeV and therefore it is sensitive to signals with $m_X \lesssim 95$ MeV.

The LO branching ratio for the decay $\mu \to eX$ can be read off from Figure 4, assuming one of the two couplings $C_{L,R} \neq 0$. We will be dealing with couplings in the range $10^{-12} \lesssim |C_{L,R}| \lesssim 10^{-9}$. Following (33), the smallness of these couplings can be related to a large scale $\Lambda$ in the range of $10^5 - 10^8$ TeV. This leads to branching ratios of the order $\mathcal{B} \sim 10^{-4} - 10^{-9}$ potentially compatible with observation at high-intensity muon facilities. As summarised in [36], this sensitivity can be competitive with other constraints on the couplings $C_{L,R}$.

Regarding the positron energy spectrum, according to (34) the LO contribution results in a delta peak for a fixed value of $E$. As we are dealing with small $m_X$, this peak is very close to the endpoint of the Michel spectrum. A more realistic description of the modification of the endpoint spectrum in the presence of $\mu \to eX$ can be obtained by a calculation at NLO in $a \equiv (\alpha/\pi)$. This leads to a radiative tail in the energy spectrum of the signal events, due to the emission of one soft photon. Hence, while a LO calculation is sufficient for the simplified analysis we will present in Section 4, we also provide a NLO calculation to prepare the theory input required for a more complete experimental analysis [92].

The NLO calculation can be done with standard techniques. The single genuine loop diagram, the vertex diagram, has infrared and ultraviolet singularities which show up as poles in

$\epsilon = (4-d)/2$. The infrared singularities are pure soft singularities and they cancel as usual when combining the virtual and real corrections. The ultraviolet singularities are absorbed by the on-shell fermion wave-function renormalisation factors and the renormalisation of the couplings $C_L$ and $C_R$. For the latter we use the $\overline{\text{MS}}$-scheme and we write the corresponding renormalised couplings at the scale $\mu_r$ as

$$\overline{C}_{L/R}(\mu_r) = \mu_r^{-2\epsilon} Z_C^{-1} C_{L/R}, \tag{36}$$

in terms of the bare couplings and the renormalisation factor $Z_C$. In what follows, a numerical value for the coupling always refers to $\overline{C}_{L/R}(M)$. Since we use the on-shell scheme for mass renormalisation and there are no internal fermion lines at tree level, no explicit mass counterterm diagrams are required.

The calculation is performed in conventional dimensional regularisation (CDR) and the four-dimensional helicity scheme (FDH) [93]. Using an anticommuting $\gamma^5$ automatically leads to the same renormalisation for $C_L$ and $C_R$ with

$$Z_C = 1 - a\,\frac{3 + n_\epsilon/2}{4\epsilon}, \tag{37}$$

where in CDR $n_\epsilon = 0$ and in FDH $n_\epsilon = 2\epsilon$. The ubiquitous factors of $\log(4\pi)$ and $\gamma_E$ associated with the pole $1/\epsilon$ are understood. Alternatively, the computation was also performed with the Breitenlohner-Maison (BM) [94] treatment of $\gamma^5$. Within the four-dimensional formulation (FDF) of FDH [95], this is a more natural choice [96]. In this case an additional finite renormalisation

$$Z_5^{\text{BM}} = 1 - a, \tag{38}$$

for the $\gamma^5$ term is required. Taking this into account together with the scheme dependence of the wave-function renormalisation a result is found that does not depend on the scheme used nor on the treatment of $\gamma^5$. As before, all analytic results are attached as an ancillary file to this submission.

While this process is simple enough to allow for an analytic calculation of the energy spectrum, we have implemented the amplitudes in MCMULE and work with numerical results. This will simplify a future full experimental analysis that may entail more involved cuts. As with the SM results, the relevant data can be obtained at the website [66].

In Figure 5 we show the functions $F$ and $G$ for the signal for various values of $m_X$, defined as in (1). In the case of a $V+A$ coupling, $F$ and $G$ at LO are identical delta functions. Due to NLO corrections, there are small differences, in particular $F$ is slightly greater than $G$ for small $E$. For a $V-A$ coupling $G$ is the same but with opposite sign, while for a pure vector or axial-vector coupling $G = 0$ (isotropic distribution).

The polarisation $P$ has an important impact on the search strategy. The background positron from polarised Michel decay has an angular dependence, as depicted in Figure 6. Depending on the nature of the ALP couplings (33) the signal has either no dependence on the angle $\theta$ (for a $V$ or $A$ coupling), a dependence similar to the background (for a $V-A$ coupling), or a dependence that is basically orthogonal to the background (for a $V+A$ coupling). In the left panel of Figure 6 we show these three extreme cases for two values of the polarisation, a realistic value of $P = -0.85$ (solid line) and a perfectly polarised muon beam (dashed lines). The ALP mass is not specified because the positron angular distribution is independent on it.

These results have been obtained by including positrons of all energies. While this does not have a significant effect on the signal, the dependence of the angular distribution of the Michel decay positron on a possible cut on the positron energy is shown in the right panel

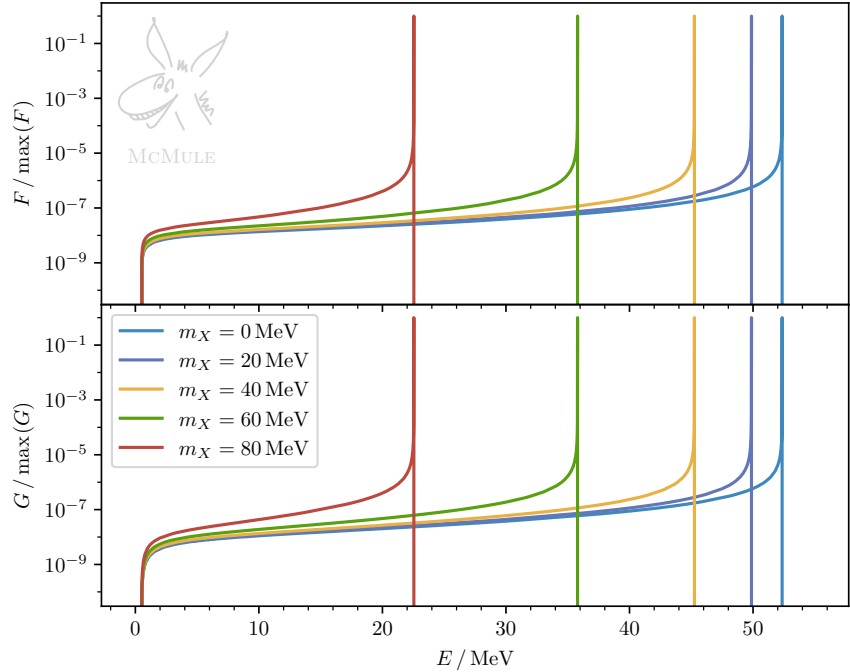

Figure 5: The functions $F$ and $G$ for $\mu \to eX$ at NLO for different masses $m_X$. While $F$ is independent of the coupling structure, $G$ is depicted for a $V+A$ coupling.

of Figure 6, again for two values of polarisation. When constraining $E$ to ever higher values, the SM distribution approaches the $V-A$ signal distribution. This is due to the kinematic configuration of the Michel decay at the endpoint. More specifically, $E$ is maximised when the two neutrinos are emitted in parallel and the positron in the opposite direction. Since two neutrinos with parallel momentum assume opposite spins, the Michel decay at the endpoint resembles a two-body decay into a positron and a scalar particle, i.e. the signal.

How the muon polarisation can be exploited to increase the signal sensitivity can be easily read from Figure 6. The search for signals with $V$, $A$ and $V+A$ coupling is enhanced in the forward region $c_\theta > 0$, especially in the latter case. The backward region $c_\theta < 0$ is instead convenient for a $V-A$ signal, especially for higher ALP masses, since the background positrons are less polarised at lower energies. Finally, it is important to obtain the highest possible muon polarisation, to further increase the angular separation between signal and background positrons.

## 3.2 LFV vector particles

It is tempting to extend the considerations to a simplified model with a light LFV vector boson, $V$. A naive approach as e.g. done in [97] leads to an apparent enhancement for small masses $m_V$. However, as we will now argue, a proper treatment of the $m_V \to 0$ limit does not have such an enhancement and, in fact, is not independent from considering scalar ALPs.

A naive simplified Lagrangian for a vector boson $V$

$$\mathcal{L}_V = V_\mu \, \bar{\psi}_j \gamma^\mu \big( C_L^{jl} P_L + C_R^{jl} P_R \big) \psi_l \,, \tag{39}$$



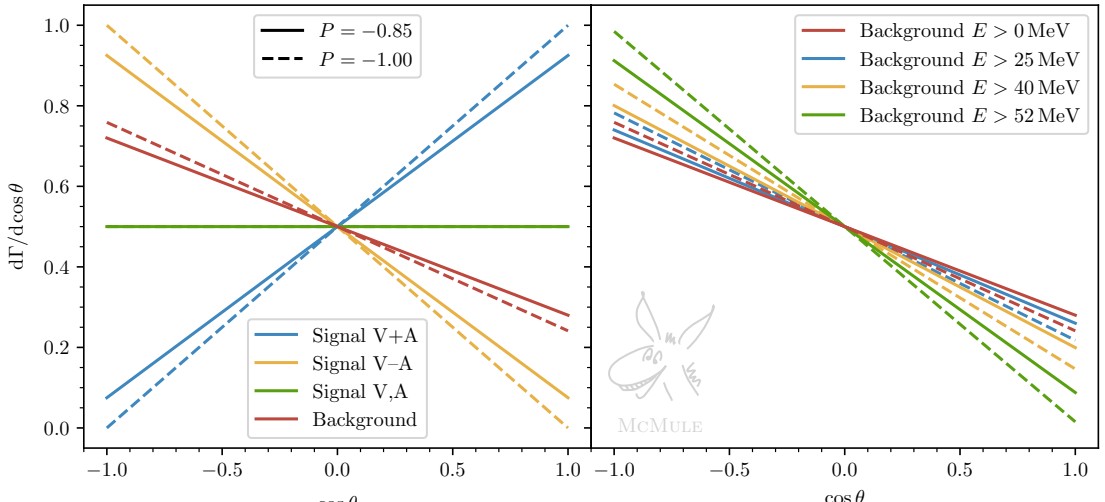

Figure 6: Left panel: The angular distribution for the signal and the background for two different muon polarisations. Right panel: The angular distribution of the background as a function of the cut on the positron energy $E$.

leads to a decay width

$$\Gamma_0^V \equiv \Gamma_0(\mu \to eV) = \frac{|\vec{q}\,|}{16\pi} \left[ \left( \frac{(1-z^2)^2}{r^2} + 1 + z^2 - 2r^2 \right) \left( |C_L|^2 + |C_R|^2 \right) - 12z \operatorname{Re}\left( C_R C_L^* \right) \right]. \quad (40)$$

This expression is divergent for $r = m_V/M \to 0$. From a technical point of view this apparent singularity appears due to the terms $p_\mu p_\nu / m_V^2$ in the polarisation sum of the vector boson with momentum $p$ and, more generically, is related to the difficulty of working with simplified models with massive vector bosons. As pointed out in [98], a more careful consideration reveals that in more complete models, $\Gamma_0^V$ is finite for $m_V \to 0$.

One possibility is to let $V$ be the gauge boson of an extra $U(1)$ gauge symmetry with coupling $g_V$ to fermions. Then, in the limit $g_V \to 0$, the mass $m_V \to 0$ as well as the couplings $C_{R,L} \to 0$, rendering $\Gamma_0^V$ finite. An alternative scenario is to consider a renormalisable model where the $\mu \to eV$ decay does not enter at tree-level, but at the loop level. In this case form factors take the place of the flavour-violating couplings in the formula for the decay rate. As shown in [98], in this situation the naively problematic $1/r$ terms always appear multiplied by form factors proportional to $m_V^2$, thus making the total of these contributions non-divergent and invariably yielding a finite rate for $\mu \to eV$ in the limit $m_V \to 0$.

Moreover, as in the case of high-energy processes with highly-boosted $W$ bosons in final states, these $1/m_V$ factors arise from the emission of the longitudinal polarisation. As is well known from the case of $W^\pm$, in the massless limit one may invoke the Goldstone boson equivalence theorem [99–102], according to which the longitudinally enhanced interactions of a massive on-shell vector $V$ can equivalently be computed by replacing it with $\partial_\mu X / m_V$, with $X$ being its corresponding Goldstone field. Hence, one can turn the generic Lagrangian (39) into an equally generic

$$\mathcal{L}_V \longrightarrow \frac{\partial_\mu X}{\Lambda} \bar{\psi}_j \gamma_\mu \left( \frac{\Lambda}{m_V} C_L^{jl} P_L + \frac{\Lambda}{m_V} C_R^{jl} P_R \right) \psi_l = \frac{\partial_\mu X}{\Lambda} \bar{\psi}_j \gamma_\mu \left( g_L^{jl} P_L + g_R^{jl} P_R \right) \psi_l, \quad (41)$$

which takes the form of (30). Therefore, for finite $g_{L,R} \propto C_{L,R}/m_V$ in the $m_V \to 0$ regime we are interested in this work, the basically massless vector behaves as the massless pseudoscalar.

Thus we refrain from performing the analysis for the former, as the conclusions taken in the previous section would require a mere translation to the parameters defined in (39).

## 4 Experimental sensitivity

In this section we estimate the expected sensitivity on the branching ratio of $\mu^+ \to e^+ X$, focusing on the impact of the theory error. We will consider three different experimental scenarios: MEG II, Mu3e, and a hypothetical forward detector. In the first two cases, we define a simplified model of the positron spectrometers of both experiments, based on their nominal geometry and expected performances. For the hypothetical forward detector, we assume different potential configurations.

Our results of this section are not to be understood as a definite answer on what limits can be obtained through a full experimental analysis. We also stress that they are not validated by the involved experimental collaborations. They are a first attempt to point out the importance of the theoretical errors and contrast them with the expected experimental errors. Even though we will show limits for a rather large range of $m_X$, we are primarily interested in the region $m_X \to 0$. In this region, i.e. at the endpoint of the positron energy spectrum, it is not possible to extract limits simply by comparing event rates in a certain signal window to event rates in the vicinity, since the background spectrum falls sharply.

In all three cases, the expected positron energy spectrum $\mathcal{P}_s$ for the signal $\mu^+ \to e^+ X$ and $\mathcal{P}_b$ for the background $\mu^+ \to e^+ \nu_e \bar{\nu}_\mu$ can be obtained from the theoretical spectrum $\mathcal{H}_{s/b}$ as

$$\mathcal{P}_{s/b}(E) = \int dE' \left[ \mathcal{H}_{s/b}(E') \times \mathcal{A}(E') \times \mathcal{S}(E, E') \right] \equiv \left( \mathcal{H}_{s/b} \times \mathcal{A} \right) \otimes \mathcal{S}, \tag{42}$$

where $\mathcal{A}$ denotes the energy acceptance function and $\mathcal{S}$ the detector response function. If the expected spectrum $\mathcal{P}_{s/b}$ is normalised, we obtain the probability density function (PDF) of the positron energy, for which we use the notation $\langle \mathcal{P}_{s/b} \rangle$.

The theoretical spectrum of the background or the signal can be obtained by integrating the corresponding decay rate, written as in (1), over the geometrical acceptance of the detector for a given muon polarisation $P$. Taking into account a detector geometry defined by the angular regions $c_m \leq c_\theta \leq c_M$ and $\phi_m \leq \phi \leq \phi_M$, we get

$$\mathcal{H}_{s/b}(E) = \frac{\phi_M - \phi_m}{2\pi} \left[ (c_M - c_m) F_{s/b}(E) - \frac{1}{2} P \left( c_M^2 - c_m^2 \right) G_{s/b}(E) \right], \tag{43}$$

where $F_b = F$ and $G_b = G$ have been evaluated in Section 2 for the background, and $F_s = F$ and $G_s = G$ have been evaluated in Section 3 for the signal. The $G_{s/b}$ contribution vanishes for symmetric geometries $c_M = -c_m$, but becomes important for forward and backward regions.

For both MEG II and Mu3e the acceptance function $\mathcal{A}$ is reasonably well described by a Gaussian cumulative distribution, given by

$$\mathcal{A}(E) = \frac{1}{\sigma_A \sqrt{2\pi}} \int_{-\infty}^{E} \exp\left[ -\frac{1}{2} \left( \frac{t - E_c}{\sigma_A} \right)^2 \right] dt = \frac{1}{2} \left[ 1 + \mathrm{erf}\left( \frac{E - E_c}{\sqrt{2}\,\sigma_A} \right) \right], \tag{44}$$

where $\mathrm{erf}(x)$ is the Gauss error function. Thus, $E_c$ corresponds to the positron energy where the acceptance is $1/2$, while $\sigma_A$ parameterises how quickly the acceptance grows from 0 to 1 for increasing $E$. The explicit values of these parameters for the three cases will be given below. For the hypothetical forward detector, we will only consider the endpoint region $E > 50$ MeV, in which we assume a constant acceptance $\mathcal{A} = 1$. In all three cases, the function $\mathcal{A}$ must be understood as the relative acceptance without overall factors, so that $\max \mathcal{A} = 1$.

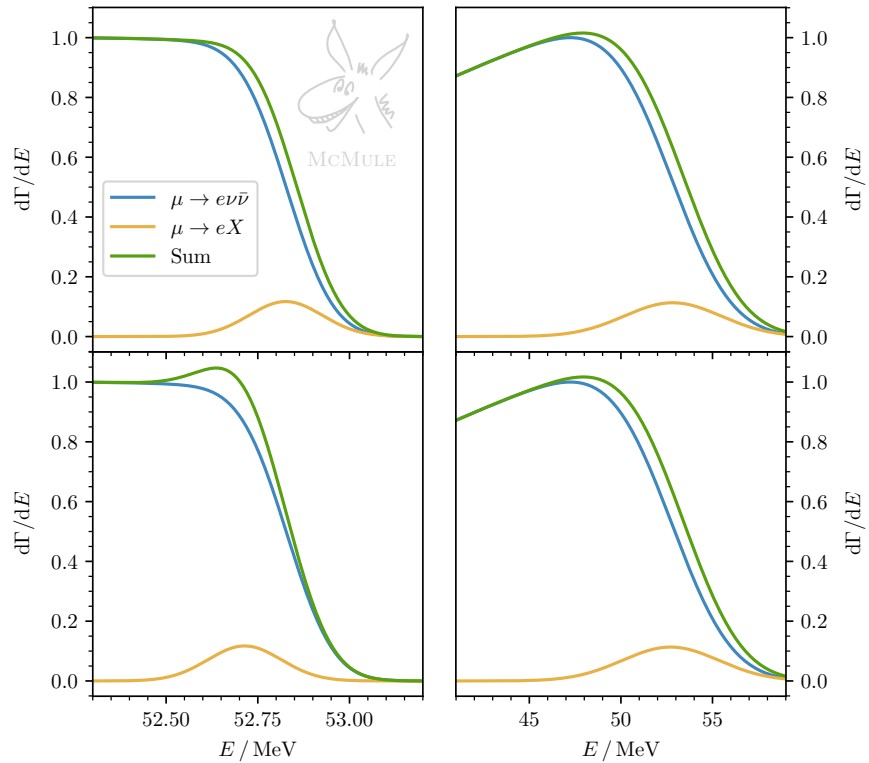

Figure 7: Comparison between signal (orange), background (blue), and combined (green) for our MEG II (left) and Mu3e (right) scenario. The branching ratio $\mathcal{B} = 5 \times 10^{-3}$ has been chosen unnaturally large to be clearly visible, with $m_X = 1\,\text{MeV}$ (top) or $m_X = 5\,\text{MeV}$ (bottom). All distributions are normalised so that the maximum of the background spectrum is 1.

In all three experimental scenarios, we parameterise the detector response function $\mathcal{S}$ with a Gaussian distribution with mean zero and a standard deviation $\sigma_S$, representing the positron energy resolution. Explicitly, we have

$$\mathcal{S}(E, E') = \frac{1}{\sigma_S(E')\sqrt{2\pi}} \exp\left[-\frac{1}{2}\left(\frac{E - E'}{\sigma_S(E')}\right)^2\right]. \tag{45}$$

A better description of the tails can be obtained with the sum of more Gaussian distributions, but this is beyond the scope of our analysis. Again, the explicit values of $\sigma_S$ for the three cases will be given below.

In order to illustrate the interplay between background and signal, in Figure 7 we consider an unnaturally large branching ratio $\mathcal{B} = 5 \times 10^{-3}$ for $\mu \to eX$, showing how it impacts the positron energy distribution near the endpoint. The parameters we use for the response function (45) and the acceptance (44) are given in Section 4.1 for MEG II (left) and Section 4.2 for Mu3e (right). For the signal events (orange) we report two values of mass, namely $m_X = 1\,\text{MeV}$ (top) and $m_X = 5\,\text{MeV}$ (bottom). The combined positron spectrum (green) is typically simply shifted with respect to the background (blue) near the endpoint. Only for sufficiently large $m_X$ and small $\sigma_S$ a peak starts to form.

The experimental sensitivity on the branching ratio $\mathcal{B}$ of $\mu^+ \to e^+X$ can be estimated by following the cut-and-count approach described in [37, 52, 103, 104]. Although a detailed

analysis will rely on more sophisticated techniques, this method gives a good indication of the role of theoretical and experimental uncertainties, particularly for small $m_X$.

As a first step, we define a signal bin. For a given value of the ALP mass $m_X$, the bin is centred at the energy $E_X$ given by (34). The bin width is $\Delta(E_X) = z_{90}\,\sigma_S(E_X)$, where $z_{90} = 1.645$ is a numerical factor depending on the choice of the confidence level (CL) and $\sigma_S(E_X)$ is the detector resolution at the bin centre. For a 90% CL the factor $z_{90}$ satisfies the equation $\mathrm{erf}(z_{90}/\sqrt{2}) = 0.9$.

The expected number of background events in this signal bin is given by

$$\mathrm{nr}_b = N_b\,I_b \equiv N_b \int_{E_X-\Delta}^{E_X+\Delta} \langle \mathcal{P}_b(E) \rangle \, dE \,, \tag{46}$$

where $N_b$ is the total number of collected background events. It is related to the number of decaying muons $N_\mu$ through the background efficiency $\mathcal{E}_b$ as $N_b = \mathcal{E}_b N_\mu$, where we have approximated the Michel decay branching ratio as 1. We define the efficiency for signal and background as the number of signal/background positrons emitted in the detector acceptance divided by the total number of signal/background positrons produced at the target, i.e.

$$\mathcal{E}_{s/b} = \frac{\int \mathcal{P}_{s/b}(E)\,dE}{\int F_{s/b}(E)\,dE} \,, \tag{47}$$

with $G_{s/b}(E)$ not being included in the denominator because its contribution vanishes when considering the total solid angle. We also define the relative signal versus background efficiency as $\mathcal{E} = \mathcal{E}_s/\mathcal{E}_b$. Since overall contributions to the detector acceptance, such as quantum and tracking efficiency, are substantially cancelled out in the ratio $\mathcal{E}$, it is not necessary to include them in (47) or in our description of the acceptance function $\mathcal{A}$.

The expected number of signal events in the signal bin depends on the $\mu^+ \to e^+ X$ branching ratio $\mathcal{B}$ as $\mathrm{nr}_s = \mathcal{B} N_\mu \mathcal{E}_s I_s = \mathcal{B} N_b \mathcal{E} I_s$, where $I_s$ is defined in analogy to (46). Using the LO approximation $\mathcal{H}_s(E') \propto \delta(E' - E_X)$, we have $I_s = \mathrm{erf}(z_{90}/\sqrt{2}) = 0.9$. At NLO the value of $I_s$ is reduced by about 1–5%, with the exact value depending on $m_X$ and $\sigma_S$. Since this correction is not required for our target precision, we use the LO approximation in our analysis.

As we will consider $N_b = 10^7 - 10^{15}$, we can approximate the bin content distribution by a Gaussian. Hence, the upper limit on the branching ratio at 90% of CL is obtained by requiring $\mathrm{nr}_s \geq z_{90}\sqrt{\mathrm{nr}_b}$ and results in

$$\mathcal{B}_{\mathrm{stat}} = \frac{z_{90}}{I_s} \frac{\sqrt{N_b\,I_b}}{N_b\,\mathcal{E}} \,. \tag{48}$$

The procedure can be repeated for any hypothesis of $m_X$. A first important correction to the previous evaluation is given by the inclusion of the theoretical error on the background. Using the error estimate of Section 2.5 we obtain the error $\Delta\mathcal{H}_b(E)$ on $\mathcal{H}_b$, given in (43). The errors on $F = F_b$ and $G = G_b$ are not independent. Near the endpoint we have $F_b \simeq -G_b$. Hence, when combining $F_b$ and $G_b$ to $\mathcal{H}_b(E) \pm \Delta\mathcal{H}_b(E)$ we treat the errors on $F_b$ and $G_b$ as completely anti-correlated. The error $\Delta\mathcal{H}_b(E)$ induces an error $\Delta\mathcal{P}_b(E)$ on the background energy spectrum through (42). In our simplified approach, we determine the average theoretical error $\delta_{\mathrm{th}}^X$ within the energy bin under consideration. This corresponds to an additional $\mathrm{nr}_b\,\delta_{\mathrm{th}}^X$ events in the bin and results in

$$\mathcal{B}_{\mathrm{th}} = \frac{I_b}{I_s\,\mathcal{E}}\,\delta_{\mathrm{th}}^X \equiv \frac{I_b}{I_s\,\mathcal{E}}\,\frac{1}{2\Delta} \int_{E_X-\Delta}^{E_X+\Delta} \frac{|\Delta\mathcal{P}_b(E)|}{\mathcal{P}_b(E)}\,dE \,, \tag{49}$$

which is the theoretical error on the branching ratio. Intuitively, in order to avoid signal biases, the number of signal events $\mathrm{nr}_s$ in the energy bin must exceed the number of events $\mathrm{nr}_b\,\delta^X_{\mathrm{th}}$ that could be due to a higher-order contribution not included in the background prediction. In order to consider the corresponding worsening of sensitivity, the theoretical error $\mathcal{B}_{\mathrm{th}}$ is added in quadrature to the statistical contribution $\mathcal{B}_{\mathrm{stat}}$. The error on $F_s$ and $G_s$ is negligible, being suppressed by the signal branching ratio. We also note that $\mathcal{B}_{\mathrm{th}}$ is the best sensitivity that an experiment can achieve in the limit $N_b \to \infty$ with the current status of the theory.

In addition to the statistical and theoretical contributions, we need to consider the presence of systematic errors in the positron reconstruction. As can be noted from Figure 7, the background spectrum with a global offset on the positron energy has the same shape of the original background with a signal at the endpoint. This results in a potential signal bias, which limits the search for small ALP masses. To include this effect on the sensitivity, we repeat the cut-and-count procedure for different PDFs by introducing an offset $\mathcal{P}_b(E) \to \mathcal{P}_b(E \pm \delta E)$ in the positron energy. The maximal variation of signal bin content, with respect to the null offset hypothesis,

$$\mathcal{B}_{\mathrm{sys}} = \frac{1}{I_s\,\mathcal{E}} \int_{E_X-\Delta}^{E_X+\Delta} \left| \langle \mathcal{P}_b(E) \rangle - \langle \mathcal{P}_b(E \pm \delta E) \rangle \right| dE \tag{50}$$

is then added in quadrature to the statistical and theoretical contributions. We mention that the main sources of systematical error, both for MEG II [103] and Mu3e [37], are the knowledge of magnetic field and muon beam polarisation, the correct alignment of sub-detectors, and the positron energy deposit in the muon target.

## 4.1 MEG II scenario

In our first toy analysis, we consider a situation inspired by the MEG II positron spectrometer [11], whose nominal angular acceptance is $|c_\theta| \le 0.35$ and $|\phi| < \pi/3$. The muons are assumed to decay at rest at the centre of the spectrometer with an initial-state polarisation of $P = -0.85$, in agreement with [105]. The parameters characterising the energy acceptance (44) are chosen as $\sigma_A = 2.5\,\mathrm{MeV}$ and $E_c = 47\,\mathrm{MeV}$ [52]. As shown in the middle panel of Figure 8, this means that only positrons near the endpoint are detected. For the energy resolution in (45), we assume the constant $\sigma_S = 0.1\,\mathrm{MeV}$ [106]. In the analysis the allowed range of $c_\theta$ is adapted to the nature of the ALP coupling, in order to increase the corresponding signal-to-background ratio. As shown in Figure 6 for $P = -0.85$, the SM positrons are preferably emitted in the backward region $c_\theta < 0$, due to the $V{-}A$ structure of Michel decay. Thus, as discussed in Section 3.1, for a $V{-}A$ coupling we limit the analysis to the region $-0.35 \le c_\theta \le 0$, whereas for a $V{+}A$, $V$ or $A$ coupling we choose the opposite direction $0 \le c_\theta \le 0.35$.

Starting with the statistical contribution, in the left panel of Figure 9 we show the corresponding upper limit (48) as a function of $m_X$ for $N_b = 10^7$ (blue), $N_b = 10^8$ (orange), and $N_b = 10^9$ (green), which is the range of positrons events that can be collected by MEG II. The size of $N_b$ is limited by the trigger selections for the $\mu^+ \to e^+\gamma$ search, especially the requirement of a photon coincidence in the LXe calorimeter. Since the majority of positron-photon coincidences are accidental and not due to a prompt $\mu^+ \to e^+ \nu_e \bar{\nu}_\mu \gamma$ decay [11], the Michel spectrum acquired by MEG II can be considered totally inclusive with respect to photon emission. Nevertheless, the effect of prompt radiative decays on the positron spectrum can be taken into account by implementing the trigger cuts in McMule. In principle, a higher number of events can be collected by relaxing the trigger conditions on the positron or even performing a dedicated run.

From a statistical point of view, we obtain better sensitivities for low ALP masses. Due to the finite resolution, the background is smaller close to the spectrum endpoint, while the signal

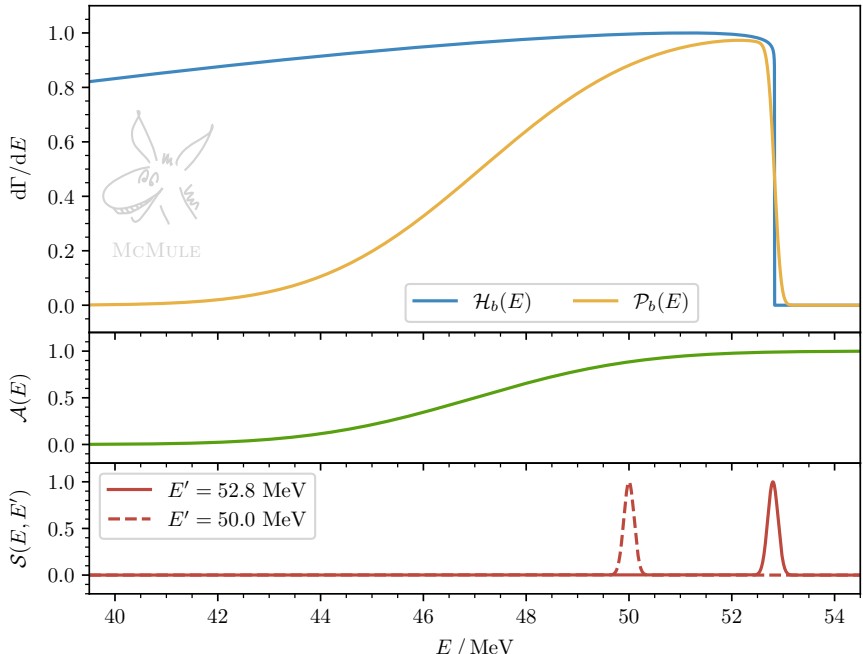

Figure 8: Expected positron energy spectrum of $\mu \to e\nu\bar{\nu}$ for MEG II (upper panel) with our assumption of experimental acceptance (middle panel) and resolution (lower panel) for $|c_\theta| < 0.35$. The theoretical spectrum $\mathcal{H}_b$ is normalised so that its maximum is 1.

acceptance is maximal. On the other hand, the sensitivity gets worse as the ALP mass increases, due to the lower acceptance of the spectrometer for low-energy positrons. For a given $N_b$, the limits are strongest for a $V+A$ interaction (solid lines), intermediate for $V$ or $A$ (dashed lines) and weakest for a $V-A$ interaction (dotted lines), due to the decreasing difference between the signal and background angular distributions. We note that MEG II can also search for signals with $m_X > 40$ MeV by lowering the intensity of its magnetic field, with the effect of increasing the positron energy acceptance. This possibility is currently being investigated [107].

The impact of the theoretical uncertainty (49) is shown in the right panel of Figure 9. Focusing on the $V+A$ case, we show the combined limit for two different errors. The effect of including our theory error (28) is shown as dashed line. Since these lines are hardly distinguishable from the solid line, representing the statistical contribution only, we refrain from including the latter in the right panel. On the other hand, if we had a NLO background prediction only, the theoretical error would make it impossible to improve the limit beyond $\mathcal{B} \sim 10^{-5}$, regardless of the statistics. This is illustrated by the dotted lines that show the combined limit using the NNLO corrections $h_2$ as theory error. The corrections beyond NLO are particularly important at the endpoint and, hence, their inclusion strongly affects the possibility to improve the limits for small $m_X$.

Unfortunately, also a systematic bias in the positron energy reconstruction has a major impact on the obtainable limits on the branching ratio, in particular for small $m_X$. This is exemplified in Figure 10, where we depict our result for $N_b = 10^9$ and various choices of misconstruction. In particular, we contrast a perfect reconstruction $\delta E = 0$ (blue curve) with two reasonable energy shifts, namely $\delta E = 2$ keV (orange) and $\delta E = 10$ keV (green). The effect of $\delta E$ is computed as described in (50). The left panel corresponds to a $V-A$ signal, while the right panel reports the $V+A$ case. The effect of the best theoretical error is included, even if negligible compared to the statistical and systematic contributions. For comparison, the limits

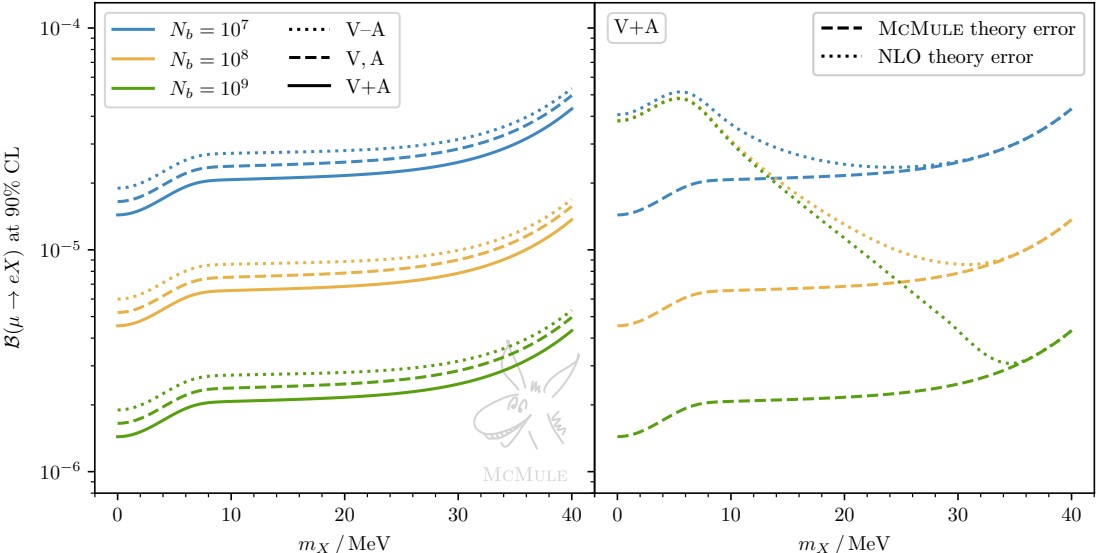

Figure 9: Sensitivity on the branching ratio for $\mu \to eX$ in our MEG II scenario. The left panel includes the statistical error only, while the right panel shows the effect of taking into account the theory error for a $V+A$ coupling. The combined limit with the best theory error (dashed line) is hardly distinguishable from the statistical contribution only. If only a NLO background prediction was available, the sensitivity (dotted lines) would be limited by theory, not statistics.

obtained by TWIST [21] are shown as a black dashed line. Since a global offset on the positron energy scale results in a false signal close to the spectrum endpoint, the systematic contribution is dramatically enhanced for small ALP masses. A rigorous control of the systematic effects on the positron energy reconstruction is therefore essential for a competitive and reliable investigation in the region $m_X < 10$ MeV. In this regard, the development of new calibration tools for the MEG II spectrometer is ongoing. One of the proposed ideas is based on the Mott scattering between a mono-energetic positron beam and the MEG II muon target [108, 109].

## 4.2 Mu3e scenario

Our second toy analysis is related to the Mu3e positron spectrometer. In this case, we chose an energy resolution depending on the positron energy as $\sigma_S = 0.05\,E$ and we specify the acceptance through $\sigma_A = 2.5$ MeV and $E_c = 15$ MeV [12]. Compared to the MEG II scenario, this leads to a better acceptance for lower positron energies, but to a worse resolution, as shown in Figure 11. More specifically, our assumption is based on the Mu3e online track reconstruction [110, 111], which is not affected by trigger selections for the $\mu^+ \to e^+e^-e^+$ search. This allows us to consider a much larger sample of events than MEG II, albeit at the expense of the single-event reconstruction accuracy. Again, we assume the muons to be polarised with $P = -0.85$ and consider a $c_\theta$ range dependent on the ALP interaction. In the case of a $V-A$ coupling we consider $-0.8 \le c_\theta \le 0$, whereas in all other cases we have $0 \le c_\theta \le 0.8$. Contrary to MEG II, there is no restriction on the azimuthal angle, i.e. $|\phi| < \pi$.

The obtainable limits on the branching ratio are shown in Figure 12. In the left panel only the statistical contribution is included, whereby we consider the cases $N_b = 10^9$ (blue), $N_b = 10^{12}$ (orange), and $N_b = 10^{15}$ (green), which is the range of positron events that can be collected by Mu3e. The sensitivity is roughly constant for $m_X < 85$ MeV, from where it starts to deteriorate due to the lowering of the spectrometer acceptance. Again, the best limits

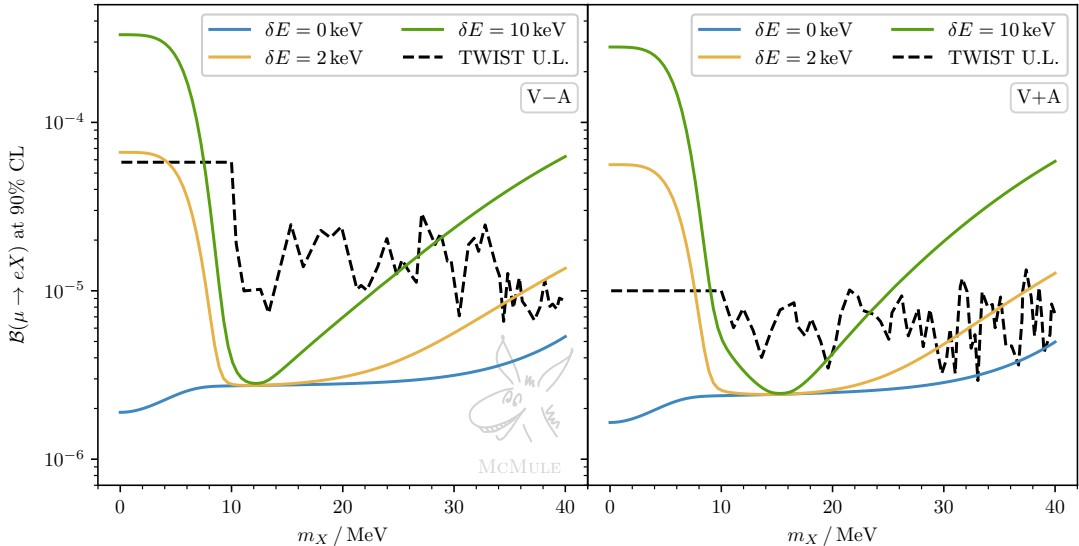

Figure 10: Sensitivity in our MEG II scenario for a $V-A$ (left panel) and a $V+A$ (right panel) signal for $N_b = 10^9$, including also a bias $\delta E$ in the positron energy reconstruction. The achievable sensitivity is strongly dependent on the systematic error, especially for $m_X < 10\,\text{MeV}$.

are obtained for a $V+A$ interaction. As shown in the right panel of Figure 6, the background positrons become less polarised as their energy decreases. Hence, the background resembles a left-handed signal near the endpoint and an isotropic signal at low energy ($E < 30\,\text{MeV}$). For this reason, the sensitivity for a $V$ or $A$ coupling becomes better than the sensitivity for a $V-A$ signal from $m_X > 60\,\text{MeV}$.

In the right panel of Figure 12, we focus on the $V+A$ case and also include the theory error. For $N_b \lesssim 10^{12}$ the improved theoretical error (dashed lines) is essential to fully exploit the statistics. Indeed, with an NLO background calculation only, the theoretical error (dotted lines) would make it pointless to increase the statistics beyond $N_b = 10^9$, as it would not lead to an improvement on the limit on the branching ratio. With our current best description of the background, the statistical and theoretical errors are about the same for $N_b = 10^{12}$. Increasing the statistics further requires further improvements in the theory description of the Michel background. Nonetheless, as already mentioned, far from the endpoint the signal appears just as a bump and the impact of the theoretical error can be reduced by comparing the number of events in different points of the spectrum. Yet, this is not possible near the endpoint, where the background spectrum falls sharply and the signal appears as a modification of the endpoint position.

As for MEG II, the calibration of the positron energy scale at the endpoint with the needed accuracy is a challenge to be specifically addressed. As discussed in [37–39], the systematic errors can be notably reduced through dedicated calibrations, based on the spectrum fit or external processes, such as Bhabha and Mott scattering.

The offline reconstruction would improve the single-event resolution to $\sigma_S = 0.1$–$0.6\,\text{MeV}$, depending on the positron energy [12]. In this way, the branching ratio sensitivity can be improved by up to one order of magnitude, although the effects due to potential trigger biases and the reduced number of events must be properly studied. Another possibility is to set the Mu3e filter farm specifically for this search, in order to increase the accuracy of the online reconstruction without sacrificing the statistics.

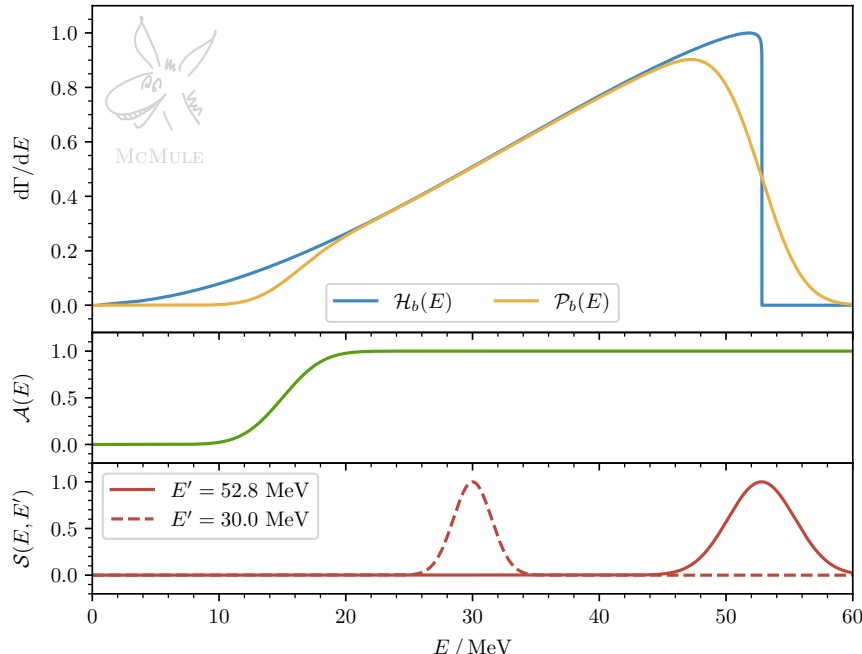

Figure 11: Expected positron energy spectrum of $\mu \to e\nu\bar{\nu}$ for Mu3e (upper panel) with our assumption of experimental acceptance (middle panel) and resolution (lower panel) for $|c_\theta| < 0.8$. The theoretical spectrum $\mathcal{H}_b$ is normalised so that its maximum is 1.

### 4.3 Forward detector scenario

The final scenario is a hypothetical forward detector, placed in the direction opposite to the muon polarisation. As shown in Figure 13, the Michel spectrum is suppressed towards the endpoint as the cut on the angular acceptance becomes more stringent, especially for a higher polarisation. Hence, this configuration is particularly well adapted to search for $V+A$ signals with $m_X \to 0$. Choosing an acceptance in the limit $c_\theta \to 1$ enhances the signal-to-background ratio, but reduces the collectable statistics in a fixed beamtime. To balance these two effects, we chose $c_\theta > 0.9$ and $|\phi| < \pi$ as a test case. In this way, assuming a rate of $10^{10}\,\mu^+/s$, as expected at HIMB [49], a sample of $10^{15}$ positrons can be collected in approximately one month of beamtime.

For our analysis, we assume a perfect polarisation $P = -1$. Although surface muons are produced fully polarised, several depolarisation effects occur during the beam production, transport, and deceleration [105]. However, it is possible to suppress the depolarisation effects by using a dedicated magnetic field to re-align the muon spin along the beam axis, as already done in [30] for this kind of search. For simplicity, we assume a constant acceptance $\mathcal{A} = 1$ for $E > 50\,\text{MeV}$ without doing any consideration for lower energies. Finally, for the energy resolution, we consider the hypotheses $\sigma_S = 0.02E$, $\sigma_S = 0.01E$, and $\sigma_S = 0.005E$, corresponding to the accuracy of typical detectors for low-energy positrons, both trackers and calorimeters.

Repeating the same analysis, we show in the left panel of Figure 14 the obtainable limits on the branching ratio including the statistical contribution only. We show the results for $N_b = 10^9$ (blue), $N_b = 10^{12}$ (orange), and $N_b = 10^{15}$ (green) with $\sigma_S = 2\%$ (dotted), $\sigma_S = 1\%$ (dashed), and $\sigma_S = 0.5\%$ (solid). As in the previous scenarios, also for the forward detector the theoretical error can be very important. To illustrate this, in the right panel of Figure 14

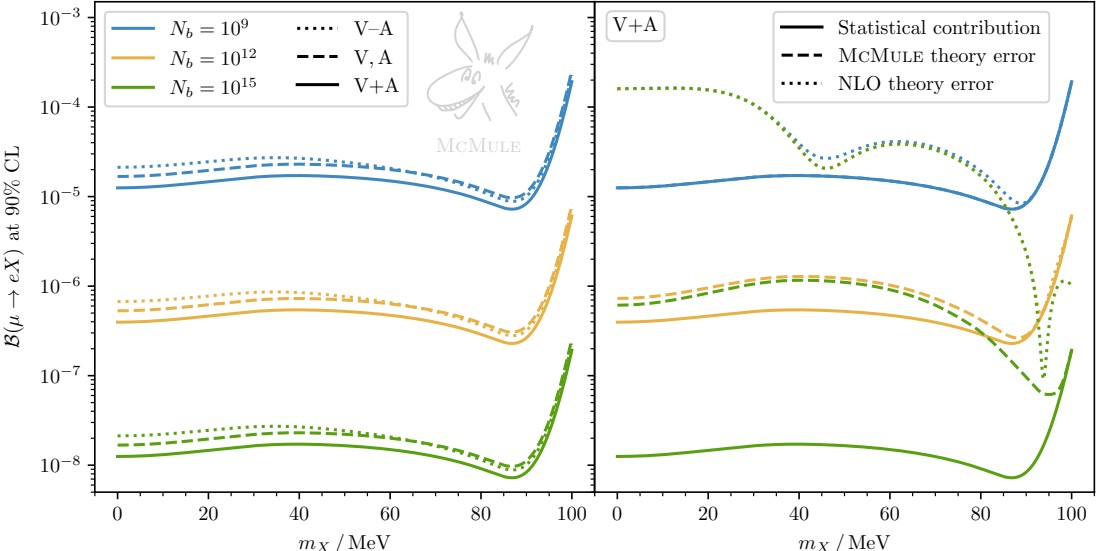

Figure 12: Sensitivity on the branching ratio for $\mu \to eX$ in our Mu3e scenario. The left panel includes the statistical error only, while the right panel shows the effect of taking into account the theory error for a $V+A$ coupling. The combined limit with the best theory error (dashed lines) is compared to the limit from the statistical contribution only (solid lines) and a NLO background prediction (dotted lines). For $N_b = 10^9$ the solid and dashed lines nearly coincide.

we compare the achievable limits for $\sigma_S = 0.005E$ in the pure statistical case (solid lines) and when the theory error is included (dashed lines). The situation is similar to the Mu3e scenario. For $N_b = 10^9$ the theory error does not affect the limit. For $N_b = 10^{12}$ the theory error is about as large as the statistical contribution, implying that further increasing the statistics to $N_b = 10^{15}$ does not lead to a substantial improvement. If only an NLO calculation was available, the theory error would limit the branching ratio to $\sim 6 \times 10^{-5}$. Thus, the corresponding curves (shown as dotted lines in Figures 9 and 12) are outside the range of Figure 14.

An appealing possibility is to use a forward detector in conjunction with the standard MEG II or Mu3e set-up, since the forward region of both experiments is not covered. A suitable detector to exploit this opportunity could be the compact calorimeter discussed in [104, 112, 113].

## 5   Conclusions

The search for a lepton-flavour-violating ALP through the modification of the Michel spectrum via $\mu^+ \to e^+ X$ is a classic case of high-intensity and high-precision frontier. Several experiments looking at rare muon decays that are ongoing or will start to take data in the near future can also consider this decay. With the forthcoming increase in the beam intensity at PSI with the HIMB project, the prospects are even better. However, to fully exploit the large statistics and the high accuracy in the measurements, the theoretical description of the background and the signal has to be known with sufficient precision. Since small $m_X$ values are the natural choice for ALPs, the endpoint of the spectrum is of special importance. This was the main motivation to reconsider the Michel decay and improve the precision in the positron energy spectrum, especially close to the endpoint. Reducing the theory error to about 5 ppm opens

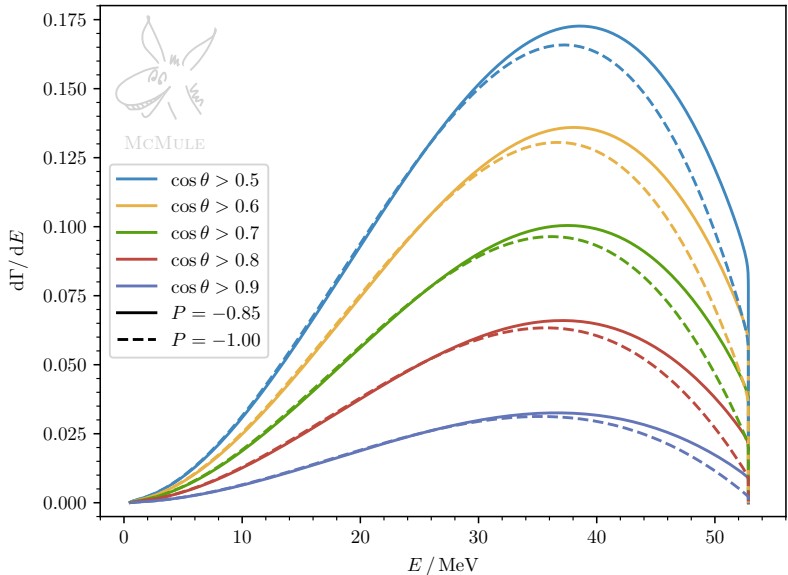

Figure 13: Theoretical positron energy spectrum of $\mu \to e\nu\bar{\nu}$ for a forward detector, computed for different geometrical acceptances and muon polarisations. The functions are normalised so that the maximum of the spectrum in the total solid angle is 1.

up the possibility to achieve limits on the branching ratio for $\mu^+ \to e^+ X$ of about $\mathcal{B} < 10^{-6}$ in a MEG II or Mu3e environment or even $\mathcal{B} < 10^{-8}$ in a dedicated experiment with a forward detector. A less careful theoretical input, using an NLO calculation only, will lead to limits that are at least two orders of magnitude worse.

The control of the systematic effects in the experiment has to match the progress in the theoretical description. In particular, a potential bias in the positron energy measurement has to be controlled very well, for example by developing dedicated calibration methods. This requires an exhaustive experimental analysis, which is beyond the scope of this paper. Nonetheless, the first steps in this direction have been taken [92], using the theoretical predictions presented in this paper to implement an improved positron event generator in the MEG II analysis software, both for $\mu^+ \to e^+ \nu_e \bar{\nu}_\mu$ and $\mu^+ \to e^+ X$.

There are other cases where the Michel decay plays an important role. In addition to being a very common background in experiments with muons, the positron spectrum endpoint is often used to calibrate low-energy detectors. Since this is one of the most basic processes in particle physics, it has served and will continue to serve as a laboratory for further progress in computational techniques. The inclusion of collinear logarithms at NNLL will become essential to further improve the theoretical prediction. Another natural next step is a full $N^3LO$ calculation of the positron energy spectrum. This would also pave the way to resum the soft logarithms at $N^3LL$. While the presence of two different non-vanishing fermion masses poses serious difficulties, it is not inconceivable that such a calculation can be done in the coming years.

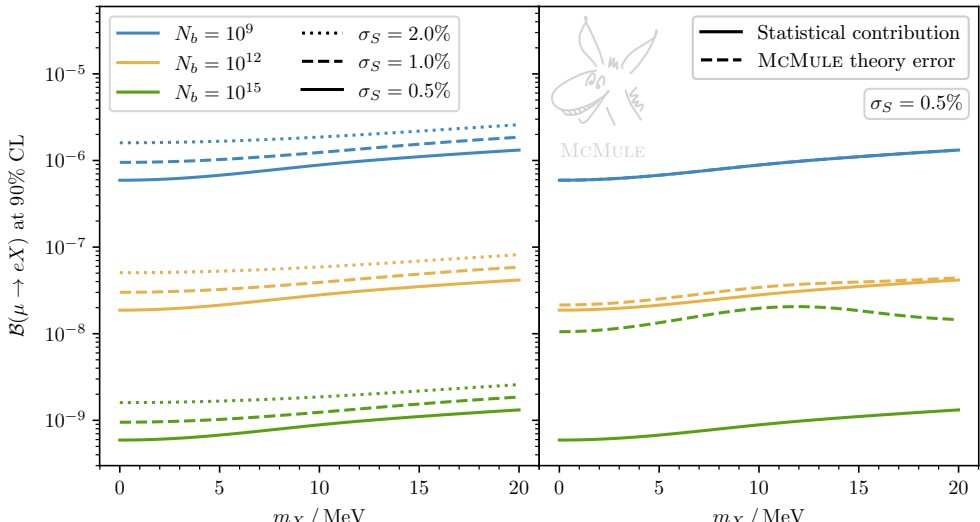

Figure 14: Sensitivity on the branching ratio for $\mu \to eX$ with $V+A$ coupling in the scenario of a forward detector with angular acceptance $c_\theta > 0.9$ and different hypothetical energy resolutions. The left panel includes the statistical contribution only, while the right panel shows the effect of taking into account the theory error for $\sigma_S = 0.5\%$. The combined limit with the best theory error (dashed lines) is compared to the limit from the statistical contribution only (solid lines). For $N_b = 10^9$ the solid and dashed lines nearly coincide.

# Acknowledgements

It is a great pleasure to thank Angela Papa and Patrick Schwendimann for collaboration at an early stage of this work. In addition, we thank Andrej Arbuzov for discussions regarding the collinear logarithms and Nik Berger, Ann-Kathrin Perrevoort, and Frederik Wauters for useful comments regarding the experimental aspects. AG is grateful to Angela Papa, Guido Montagna, and Fulvio Piccinini for their support.

**Funding information**   This work was supported by the Swiss National Science Foundation under contract 178967 and 207386. PB acknowledges support by the European Union's Horizon 2020 research and innovation programme under the Marie Skłodowska-Curie grant agreement No. 701647. AMC acknowledges support by MCIN/AEI/10.13039/501100011033, Grant No. PID2020-114473GB-I00, and by the Generalitat Valenciana, Grant No. PROME-TEO/2021/071.

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
