# Peer review of "High-precision muon decay predictions for ALP searches"

_SciPost Physics, doi:SciPost Phys. 15, 021 (2023)_

## Round 1 · Referee Report · Anonymous (Referee 1) · 2022-12-15

Report

In this important paper, theoretical description of the polarized muon
decay is improved. Results are usefully presented in an electronic
form (Python notebooks) on the McMule collaboration web page
(Ref. [64]). Applications to new physics searches are discussed in the
context of ongoing and near-future experiments.
I have only one question/suggestion: In Caola et al., ``Muon decay
spin asymmetry,'' Phys. Rev. D90, 053004 (2014), arXiv:1403.3386, the
spin asymmetry of the muon decay was computed through O(alpha^2). Does
this offer any useful cross-checks of the present results? (If not,
feel free to ignore it.) It seems that the issue of treating the
open-lepton production, noted in Section 2.1 of the present paper, was
also encountered there.
The authors correctly point out that the muon decay continues to
provide a useful test-case for new theoretical techniques. I recommend
this paper for publication.
  • validity: -
  • significance: -
  • originality: -
  • clarity: -
  • formatting: -
  • grammar: -

Author:  Adrian Signer  on 2023-02-27  [id 3409]

(in reply to Report 1 on 2022-12-15)

We thank the referee for the appreciation of our paper and for bringing up the muon spin asymmetry.

Regarding the muon spin asymmetry and the article 1403.3386, comparison is not directly possible. Not surprisingly, at NLO we agree perfectly. However, beyond NLO there are small differences. The work in 1403.3386 uses massless electrons. Hence, in their case open lepton production is not independently infrared safe and has to be combined with Michel decay. Furthermore, they have to introduce electron jets, depending on a resolution parameter (denoted by y in their paper).

We use massive electrons and open lepton production can in principle be considered completely independently of the normal Michel decay.

If we use a resolution parameter y ~ mass of the electron, the NNLO coefficient of the spin asymmetry in 1403.3386 is within 10%-20% of our result. But we stress again that we are comparing different quantities.

What is common though is that a precise description is needed on how to treat events with two positrons. We have added a remark and mention the paper 1403.3386 before (1), when discussing Michel decay vs. open lepton production.

---

## Round 1 · Referee Report · Anonymous (Referee 2) · 2022-12-23

Report

The authors present the theoretical prediction of the positron energy spectrum for the polarised muon decay process, which is of interest to several experimental studies, including the MEG II and Mu3e experiments. The paper relies strongly on earlier results and thus scores relatively low on the criterium of novelty.

One of the aspects of the paper is the inclusion of next-to-next-to-leading order corrections and logarithmically enhanced terms at even higher orders in the electromagnetic coupling constant. All these terms have been known before for the Michel spectrum.

In terms of applications, the authors apply their results to the context of the MEG II and Mu3e experiments to estimate the impact of the theory error on the branching ratio sensitivity for the lepton-flavor-violating decay of a muon into an axion-like particle. While this is a valuable application, it would be helpful if the authors could provide more context and discuss the relevance of this result also for other experiments searching for signals of charged lepton flavor violation.

Overall, this solid paper contributes to the field of theoretical particle physics. However, there are a few areas where the presentation could be improved. With some revisions, the article can be accepted. I have the following comments for the authors to consider: 1) Eq. 1 is not well defined beyond NLO on the theory side and, consequently, not at all experimentally. The issue should be clearly discussed here, or one should refrain from introducing eq. 1 so early in the paper. The comment in section 2.1 on page 4 needs to be more satisfactory and should provide more details. 2) Why is the relative uncertainty of the constant term in eq. 21 so large? The authors could comment on this issue. 3) What is the impact of f^{ee}_2 for energies lower than 40 MeV? 4) What is the physical reason for the larger impact of the collinear logarithms near the endpoint? Is it also true for the muon spin asymmetry? 5) The procedure of defining theoretical error is very ad-hoc. How should the error defined in eq 28 be interpreted in the statistical sense? How should it be combined with the experimental uncertainty? 6) It could be mentioned that the equivalency of eq. 30 and eq. 31 requires that vector and axial currents are not anomalous. 7) I need clarification on section 3.2. Instead of overstretching the applicability of their computations, the authors could compute the corrections directly for a vector case in a proper effective field theory framework. Otherwise, this section can be removed from the paper. 8) I also recommend improving the overall quality of graphics. Plots should not overlap with the legend box (see fig 1 second panel) or the logo (figure 9) 9) Finally, I could not install and test the code accompanying the submission. I followed the link to https://gitlab.com/mule-tools/mcmule and then the instruction git clone --recursive https://gitlab.com/mule-tools/monte-carlo, which prompted the need to login to git to access the files. It seems they were not made publicly available, as even logging in, I could not access the files. Fortunately, data provided in https://mule-tools.gitlab.io/user-library/michel-decay/f-and-g/ were easily accessible.

  • validity: high
  • significance: ok
  • originality: low
  • clarity: ok
  • formatting: acceptable
  • grammar: excellent

Author:  Adrian Signer  on 2023-02-27  [id 3408]

(in reply to Report 2 on 2022-12-23)

We thank the referee for the detailed comments which helped us to improve our paper. In addition to a hopefully clearer presentation we also eliminated a minor bug in the code that did, however, not affect the final results (see point 2 below). Furthermore, we have added some citations.

First we would like to address the general comments of the referee at the beginning of the report, regarding novelty, relevance, and further context.

The referee states "All these terms have been known before for the Michel spectrum.". This is not true for the NNLL soft logarithms. These logarithms strongly affect the endpoint of the spectrum and, hence, their inclusion is absolutely mandatory for nearly massless ALPs. Accordingly, we respectfully disagree with the statement "scores relatively low on the criterium of novelty". We have presented new analyses that are now possible. They simply would not be, without the NNLL improvement.

It is a unfortunate fact that in many theory papers describing the high-precision search for low-mass new physics the theory error from the Standard Model background is completely ignored. Giving meaning to such searches and potentially open the possibility to actually do them and obtain better limits does, in our view, not score low on the criterion of novelty.

We accept that this aspect has not been made clear enough in the text. Accordingly, we have added remarks in the introduction (after citing [50,51]) and at the beginning of Section 2.3.

Regarding "further context", our computation is related to a dedicated high-precision determination of the endpoint of the positron energy spectrum in the muon (or in principle tau) decay. As such, it is mainly relevant for experiments with outstanding precision and accuracy. In practice, this means for the time being it is related solely to MEG and Mu3e. The precision in the spectrum of tau decays is nowhere near precise enough to justify the inclusion of our corrections. Since the referee in a different context (point 7) asks us not to overstretch the applicability of our computations, we refrain from adding anything about the tau.

We now give our reply on the specific points raised by the referee.

1) We have separated even more clearly the discussion of (what we call) the standard Michel decay and (what we call) the open lepton production. To this end we have moved forward the mention of the two processes, just before (1). Furthermore, we have extended the discussion of events with more than one positron in the final state in Section 2.1.

2) Since no analytic result is available for the NNLO energy spectrum, this constant is extracted from numerical results at NNLO. Furthermore, the impact of the relatively large error on this constant is negligible in the total error. Hence, no particular effort is made to determine it more precisely. We have added a comment about this in the text after (21). The bug mentioned above resulted in a change of this constant from -5 +/- 1 to -6 +/- 1 which, however, has no notable effect.

3) The impact of f^{ee}_2 and g^{ee}_2 are shown in Figures 1 and 2 (blue curve, 3rd panel from the top) for the complete energy range.

4) Looking at the panel second from bottom of Figures 1 and 2, the collinear logarithms are not particularly enhanced near the endpoint. It is true that for the case of G (Figure 2) there are bumps near the endpoint, but the LO result is also large (in magnitude) there. As a result of these bumps the error due to collinear and soft logarithms are of the same size for G. This is the main reason for including the factor \sqrt{2} in (28).

5) We strongly disagree with the term 'ad-hoc'. The error has three components as indicated in (28). We suppose the referee's comment regards the terms in curly brackets. This is a careful and conservative estimate of the impact on non-calculated higher-order terms. It is a systematic theory error and has no statistical interpretation. As with every such estimate, there is no unique prescription. For each component we take as the error the last contribution that has been reliably computed. It is much more sophisticated and reliable than the widely used variation of a scale by a factor of 2.

We have completely rewritten the text after (28) to describe better our approach to estimate the impact of higher-order terms that are not included in our results.

The question "How should it be combined with the experimental uncertainty?" is explained in great detail in Section 4, which starts with the sentence "In this section we estimate the expected sensitivity on the branching ratio of µ+ → e+ X, focusing on the impact of the theory error."

6) We thank the referee for this suggestion. A corresponding remark has been added just before (31).

7) We are working in a simplified-model approach, suitable for low-mass ALPs. We do not see how we could use an EFT approach in this case. The main objective of Section 3.2 is to make the point that a naive extension to vector particles, i.e. a naive simplified-model approach, is not appropriate (even though it is done in the literature). While this is a negative statement, we think it is important to include it for the sake of completeness. Thus, we do not want to "overstretch" the applicability of our analysis. Quite to the contrary, we want to restrict it. We have made this more clear with a statement at the beginning of Section 3.2.

8) We appreciate that Figures 1 and 2 are busy. We have remade them to avoid any overlap between the legends and the curves. In each figure we have taken care that labels, legends and other elements were large enough to be easily readable. Regarding the logo, it is implemented as a watermark and does not affect 'readability' of the curves. Whether or not it is done like this is a matter of taste and has not created any issues in several previous articles.

9) We thank the referee for pointing this out. The URL was indeed incorrect due to a now rectified internal procedure related to code releases. The current URL is now visible on the website. The correct command is: git clone --recursive https://gitlab.com/mule-tools/mcmule

---

## Round 2 · Referee Report · Anonymous (Referee 1) · 2023-3-13

Report

The second version adequately addresses all the issues arising from the referee reviews. I recommend it for publication.

---

## Round 2 · Referee Report · Anonymous (Referee 2) · 2023-3-28

Report

I want to thank the authors for their careful and detailed responses to my comments. I was also able to install and run the provided code successfully. The modified version of the paper and the authors' comments provide a clear and sufficient answer to my remarks, and the paper meets the criteria for acceptance. In particular, I agree that uncertainty is reliable for practical applications.

---

## Round 2 · Author Response

We have revised the submission following guidance from the reviewers.

---

## Round 2 · List of Changes

The changes we have made as a response to the reviewer's comments are listed in our replies to the report.

---

## Editorial Decision

published